

# Towards a Bulk Approach of Local Interactions of Hydrometeors

Manuel Baumgartner[1,2] and Peter Spichtinger[1]

[1]Institute for Atmospheric Physics, Johannes Gutenberg University, Johann-Joachim-Becher-Weg 21, 55128 Mainz, Germany
[2]Zentrum für Datenverarbeitung, Johannes Gutenberg University, Mainz, Germany

*Correspondence to:* Manuel Baumgartner (manuel.baumgartner@uni-mainz.de)

**Abstract.** Growth of small cloud droplets and ice crystals is dominated by diffusion of water vapour. Usually, Maxwell's approach of growth for isolated particles is used in describing this process. However, recent investigations show that local interactions between particles can change diffusion properties of cloud particles. In this study we develop an approach for including these local interactions into a bulk model approach. For this purpose, a simplified framework of local interaction is proposed and governing equations are derived from this setup. The new model is tested against direct simulations and incorporated into a parcel model framework. Using the parcel model, possible implications of the new model approach on clouds are investigated. The results indicate that for specific scenarios the lifetime of cloud droplets in subsaturated air may be longer; these effects might have impact on mixed-phase clouds, e.g. in terms of riming efficiencies.

## 1 Introduction

Only recently, spatial distribution of hydrometeors, i.e. cloud droplets and ice crystals, has attained great attention in the context of small-scale turbulence in clouds. From idealized numerical simulations as well as experiments in cloud chambers one realized that hydrometeors may cluster in some regions of the clouds while other regions are relatively void (Shaw et al., 1998; Wood et al., 2005). Clustered hydrometeors influence each other in their diffusional growth by modifying the local field of supersaturation (see Castellano and Avila, 2011, for the case of droplet clusters). Although the existence of such clusters in real clouds remains quite controversial at the moment (Kostinski and Shaw, 2001; Vaillancourt and Yau, 2000; Devenish et al., 2012), it raises the question about their importance on the evolution of a whole cloud. The studies by Vaillancourt et al. (2001) and Vaillancourt et al. (2002) argue from direct numerical simulations, that local effects due to clustered cloud droplets in warm clouds may indeed modify diffusional growth of individual droplets but are not visible in the overall droplet spectrum. For typical turbulent situations occurring at cloud edge, Celani et al. (2005) and Celani et al. (2007) found much stronger influences of the local effects, although they probably excluded the mean growth of the droplets (Grabowski and Wang, 2013). However, the treatment of diffusional growth in all numerical cloud models relies on the diffusional growth theory developed by Maxwell and therefore assumes nearby hydrometeors not to affect each other regarding their diffusional growth behavior (see for example Rogers and Yau, 1989; Lamb and Verlinde, 2011; Wang, 2013; Maxwell, 1877).

In a mixed-phase cloud the picture may change dramatically since an ice crystal has a much severe impact on the droplets in its vicinity: it may evaporate nearby droplets and grow at their expense. This local interaction is not new and commonly referred to as the Wegener-Bergeron-Findeisen process (Wegener, 1911; Bergeron, 1949; Findeisen, 1938). Although this process is by




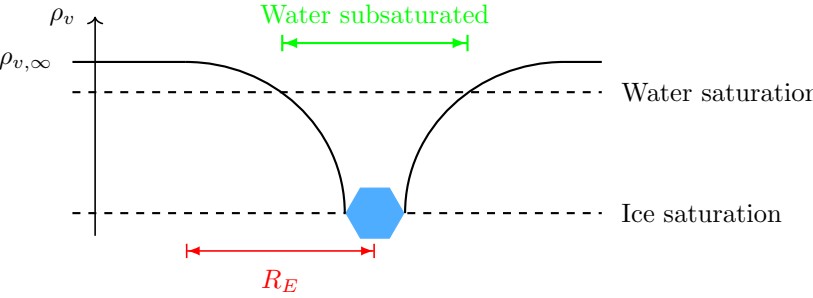

**(a)** Water supersaturated environment

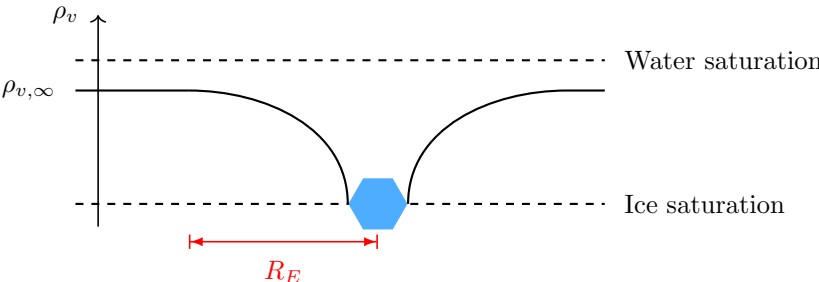

**(b)** Water subsaturated and ice supersaturated environment

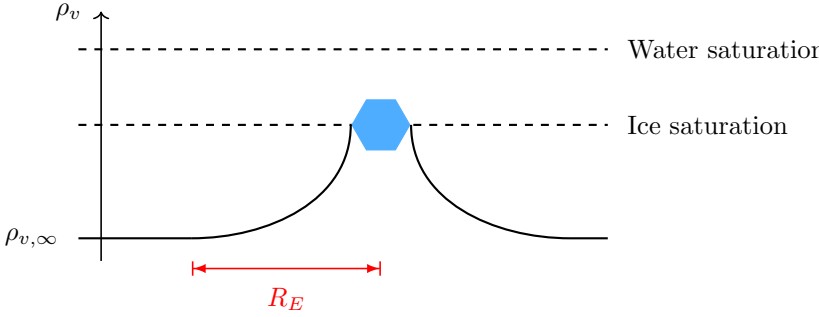

**(c)** Ice subsaturated environment

**Figure 1.** Schematic of the water vapor density field about an ice crystal with various environmental water vapor values $\rho_{v,\infty}$. Water vapor density at the ice crystal surface is assumed to equal the ice saturation density. In all cases, the ice crystal deforms the water vapor field locally within a ball of radius $R_E$. If a droplet has distance smaller than $R_E$ from the ice crystal, its growth behavior is determined by the local value of water density instead of the environmental value $\rho_{v,\infty}$.



definition a local process, numerical models do not represent it as such, since Maxwell's theory is applied and all hydrometeors grow according to the environmental rather than the local conditions. In Baumgartner and Spichtinger (2017b), we investigated the impacts of local interactions by diffusion between an ice crystal and surrounding droplets qualitatively using a reference model which resolves the hydrometeors as well as the vapor and temperature fields. For convenience, we put the results from

Baumgartner and Spichtinger (2017b) into a "guiding schematic", illustrating the local water vapor field configurations, see figure 1. If a droplet has distance smaller than $R_E$ from the ice crystal, only the local value of water vapor density is seen by the droplet and therefore dictates its diffusional growth behavior. If the local droplet number is high enough, they influence the ice crystal and may even disconnect its growth behavior from the environmental conditions. In both cases, Maxwellian growth theory is not applicable. Apart from numerical simulations, the question of solvability of the underlying governing equations is

addressed in Baumgartner and Spichtinger (2017a); for a reduced model problem existence and uniqueness of solutions could be proven.

In this study, we focus on a theoretical description of local interactions between an ice crystal and nearby droplets suited to be incorporated into a bulk-microphysical formulation. This study is organized as follows: section 2 contains a derivation and a discussion of the model equations. In section 3 we describe the incorporation of the new model into a simple parcel model in

order to assess possible implications on a whole cloud. Finally we end with a summary and concluding remarks in section 4.

## 2   Derivation of the Model Equations

This section is dedicated to the derivation of the model equations describing local interactions between an ice particle and surrounding cloud droplets. Subsection 2.1 contains the derivation. We comment on the choice of the involved parameters in subsection 2.2 and afterwards present a model simplification in subsection 2.3. Subsection 2.4 contains a general discussion of

the suggested model equations.

### 2.1   Local Ice-Droplet System

As a conceptual model for a local configuration of an ice crystal and a droplet, we consider the schematic shown in figure 2. In the following, we always assume the ice crystal as spherical, because even for non-spherical ice crystals, in some distance the shapes of the water vapor and temperature fields are very similar to the fields of a spherical ice crystal (Lamb and Ver-

linde, 2011, figure 8.24 and the analysis in Baumgartner and Spichtinger, 2017b). The ice crystal with radius $R_i$ is located in the center. Water vapor density and temperature at the surface of the ice crystal are denoted by $\rho_{v,i}, T_i$, respectively. The droplet with radius $r_d$ is located at distance $R_d$ from the ice crystal center. We henceforth call $R_d$ the "coupling distance" of the ice crystal–droplet interaction. Let $\rho_{v,d}, T_d$ denote water vapor density and temperature at the surface of the droplet, respectively. Radius $R_E$ denotes the "radius of influence" of the ice crystal, defined as the radius, where the ice crystal deforms

the surrounding fields of water vapor and temperature in a non-negligible manner. A radius of influence $r_E$ for the droplet is defined analogously. Those radii give rise to a "sphere of influence" about the respective hydrometeor, wherein their influence on the fields for water vapor and temperature cannot be ignored. We discuss possible choices of the radii of influences for both





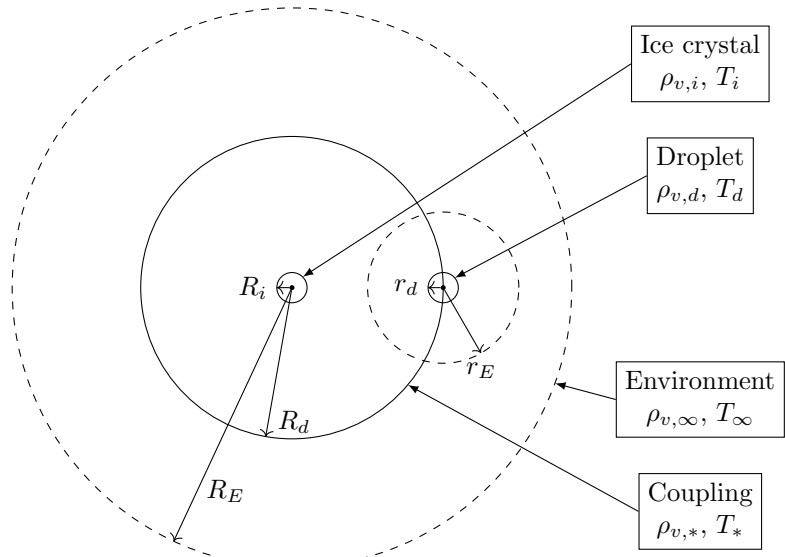

**Figure 2.** Conceptual model of local ice-droplet system. In the center is an ice crystal with radius $R_i$ and at distance $R_d$ from the ice crystal is a cloud droplet with radius $r_d$. The presence of the ice crystal and the droplet give rise to influence spheres with radii $R_E$ and $r_E$, respectively. Distance $R_d$ is the coupling distance, where the diffusional growth of the hydrometeors is coupled.

hydrometeor types in section 2.2, but objective estimates of $R_E$ and $r_E$ are not possible. The values for water vapor density and temperature at the boundary of the influence sphere about the ice crystal are given by the environmental values $\rho_{v,\infty}$ and $T_\infty$. The corresponding values at coupling distance $R_d$ are denoted by $\rho_{v,*}$ and $T_*$. We will describe the local coupling of both hydrometeors with the help of the values $\rho_{v,*}, T_*$, so we call them "coupling values". Within this study, we denote variables

referring to properties of the ice crystal with uppercase letters and variables referring to the droplet with lowercase letters.

In the following, we consider water vapor and temperature fields as spherically symmetric inside the respective influence spheres as is done in Maxwellian growth theory. This assumption is not fully consistent with the schematic in figure 2, since spherical symmetry is not able to account for a spatial localized droplet as depicted in the schematic. Assuming spherical symmetric fields means that the droplet is smeared over the sphere with radius $R_d$ and serves as a continuous source or sink for water vapor and temperature.

This will allow the generalization to the case of multiple droplets. We will use the coupling values $\rho_{v,*}$ and $T_*$ as environmental values for droplet growth, i.e. as values at distance $r_E$ from the droplet. This amounts to bloat the sphere with coupling radius $R_d$ into an artificial spherical shell with size $2r_E$ and a uniform distribution of water vapor and temperature inside this shell, compare figure 3.

However, the idea is as follows. The ice crystal first establishes fields for water vapor and temperature as if it were isolated,

yielding values at the coupling distance $R_d$. These fields are given by the equilibrium fields for water vapor and temperature about the ice crystal and serve as background fields, see equation (1). Since the droplet is located at the coupling distance $R_d$, the coupling values $\rho_{v,*}, T_*$ define the ambient values for its diffusional growth, i.e. the values at distance $r_E$ from the droplet.



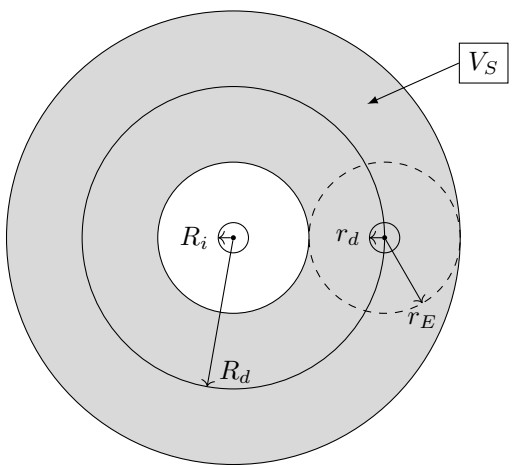

**Figure 3.** Illustration of the artificial spherical shell, obtained by "bloating" the surface of the sphere with interaction radius $R_d$ about the ice crystal. Within the gray shaded area, representing the volume $V_S$ of the artificial spherical shell, we assume a uniform distribution of water vapor and temperature.

We establish an equation for the coupling values (see equations (6) and (5)), causing the droplet to grow or evaporate. This growth or evaporation feeds back to the coupling values and will in turn influence the growth behavior of the ice crystal.

The spherical symmetric solutions of the Laplace equations $\Delta\rho_v = 0$ and $\Delta T = 0$ for $\rho_v$ and $T$ inside the influence sphere of the ice crystal, describing the water vapor density and temperature field, are given by

$$\rho_v(R) = \frac{\rho_{v,\infty}R_E - \rho_{v,i}R_i}{R_E - R_i} + \frac{(\rho_{v,i} - \rho_{v,\infty})R_iR_E}{R_E - R_i}\frac{1}{R}, \tag{1a}$$

$$T(R) = \frac{T_\infty R_E - T_iR_i}{R_E - R_i} + \frac{(T_i - T_\infty)R_iR_E}{R_E - R_i}\frac{1}{R} \tag{1b}$$

where $R$ is the radial distance from the ice crystal center. Equations (1) define the unperturbed background fields for an isolated ice crystal, representing the spatial thermodynamic equilibrium. The corresponding solutions inside the influence sphere of the droplet are

$$\widehat{\rho_v}(r) = \frac{\rho_{v,*}r_E - \rho_{v,d}r_d}{r_E - r_d} + \frac{(\rho_{v,d} - \rho_{v,*})r_Er_d}{r_E - r_d}\frac{1}{r}, \tag{2a}$$

$$\widehat{T}(r) = \frac{T_*r_E - T_dr_d}{r_E - r_d} + \frac{(T_d - T_*)r_Er_d}{r_E - r_d}\frac{1}{r} \tag{2b}$$

where $r$ is the radial distance from the droplet center. Proceeding as in the derivation of the Maxwellian growth equations, we obtain from (2) the following equations for the change in droplet mass and temperature

$$\frac{dm_d}{dt} = 4\pi\alpha_d D_0 r_d r_E \frac{\rho_{v,*} - \rho_{v,d}}{r_E - r_d}, \tag{3a}$$

$$\frac{dT_d}{dt} = \frac{4\pi r_d r_E}{m_d c_{p,l}(T_d)(r_E - r_d)}\Big(L_{lv}(T_d)\alpha_d D_0(\rho_{v,*} - \rho_{v,d}) + K_0(T_* - T_d)\Big), \tag{3b}$$





where $m_d$ denotes the droplet mass, $T_d$ the droplet temperature, $\alpha_d$ the accommodation coefficient (Davis, 2006), $D_0$ the diffusivity of air, $K_0$ the thermal conductivity of air, $c_{p,l}$ the specific heat capacity of liquid water and $L_{lv}$ the latent heat of vaporization. Likewise, the growth equations for the ice crystal are given by

$$\frac{\mathrm{d}M_i}{\mathrm{d}t} = 4\pi\alpha_i D_0 R_i R_d \frac{\rho_{v,*} - \rho_{v,i}}{R_d - R_i},$$  (4a)

$$\frac{\mathrm{d}T_i}{\mathrm{d}t} = \frac{4\pi R_i R_d}{M_i c_{p,i}(T_i)(R_d - R_i)}\Big(\alpha_i D_0 L_{iv}(T_i)(\rho_{v,*} - \rho_{v,i}) + K_0(T_* - T_i)\Big)$$  (4b)

where $M_i$ denotes the ice crystal mass, $T_i$ the ice crystal temperature, $\alpha_i$ the accommodation coefficient, $c_{p,i}$ the specific heat of ice and $L_{iv}$ the latent heat for sublimation. Note that in equations (3), (4) the coupling values $\rho_{v,*}$, $T_*$ show up to allow the coupling of the diffusional growth of the hydrometeors.

After the description of the respective growth equations, we proceed to the coupling values. We define

$$T_*(t) := T(R_d) \quad \text{and consequently} \quad \frac{\mathrm{d}T_*}{\mathrm{d}t} = \frac{\mathrm{d}}{\mathrm{d}t}\Big(T(R_d)\Big),$$  (5)

with $T$ being the unperturbed background temperature field of the ice crystal from equation (1). As in Maxwellian growth theory, in definition (5), we assume temperature fluctuations to balance quickly and not to affect the droplet growth. We define the evolution equation for the coupling value $\rho_{v,*}$ as

$$\frac{\mathrm{d}\rho_{v,*}}{\mathrm{d}t} := I_1 + I_2 + I_3$$  (6)

being the sum of three terms $I_1$, $I_2$, $I_3$, representing three different processes.

The first term $I_1$ takes the unperturbed background value $\rho_v(R_d)$ of an isolated ice crystal into account. As rate of change for this value, we define

$$I_1 := \frac{\mathrm{d}}{\mathrm{d}t}\Big(\rho_v(R_d)\Big).$$  (7)

If no droplet is present, we will have $I_2 = I_3 = 0$ in equation (6) and definition (7) simply ensures $\rho_{v,*}$ to equal the background value $\rho_v(R_d)$.

The second term $I_2$ considers the change of water vapor density due to droplet growth or evaporation. We first calculate the rate $J_d$ of water vapor exchange through the surface of the ball of influence $b_E$ around the droplet. Using representation (2) of the water vapor field, we obtain

$$J_d = -\int_{\partial b_E} D_0 \nabla\widehat{\rho_v} \cdot N \,\mathrm{d}\sigma = -D_0 \frac{\rho_{v,*} - \rho_{v,d}}{r_E - r_d}\frac{r_d}{r_E}4\pi r_E^2 = -\frac{1}{\alpha_d}\frac{\mathrm{d}m_d}{\mathrm{d}t}$$  (8)

where $N$ denotes the outer normal vector at the surface $\partial b_E$. The released water vapor is assumed to diffuse in unit time into some volume $V$, to be defined later, leading to the water vapor exchange rate $\frac{J_d}{V}$. Due to the assumed spherical symmetry, using the exchange rate $\frac{J_d}{V}$ directly amounts to assuming the artificial spherical shell around the ice crystal as being filled up with droplets and their influence spheres, resulting in a strong overestimation of the droplet effect, so we have to rescale this rate. Let

$$Z := \frac{\frac{4}{3}\pi\big((R_d + r_E)^3 - (R_d - r_E)^3\big)}{\frac{4}{3}\pi r_E^3} = \frac{(R_d + r_E)^3 - (R_d - r_E)^3}{r_E^3}$$  (9)





measuring the number of droplet influence spheres fitting inside the artificial spherical shell around the ice crystal. The new value of $I_2$ is given by $\frac{J_d}{ZV}$, taking the smearing-effect into account. If we consider not only a single droplet but in total $N_d$ droplets inside the influence sphere of the ice crystal, we finally define the exchange rate as

$$I_2 := N_d \frac{J_d}{ZV} = -\frac{N_d}{\alpha_d ZV} \frac{\mathrm{d}m_d}{\mathrm{d}t}, \tag{10}$$

where we used equation (8). This rate actually neglects local interactions between the droplets inside the influence sphere and assumes all droplets as identical.

Finally, the third term $I_3$ accounts for the relaxation of the coupling value for water vapor density $\rho_{v,*}$ to $\rho_v(R_d)$, provided by the background field and representing thermodynamic equilibrium. Motivated by Fick's law of diffusion, let

$$-\int_{\partial B_d} D_0 \frac{\rho_{v,*} - \rho_v(R_d)}{R_E - R_d}\,\mathrm{d}\sigma = -4\pi R_d^2 D_0 \frac{\rho_{v,*} - \rho_v(R_d)}{R_E - R_d} \tag{11}$$

be the rate of water vapor exchange from ball $B_d$ to the outside, where $B_d$ denotes the ball with coupling radius $R_d$ about the ice crystal. The change of the coupling vapor density is therefore given by

$$I_3 := -\int_{\partial B_d} D_0 \frac{\rho_{v,*} - \rho_v(R_d)}{R_E - R_d}\,\mathrm{d}\sigma\, \frac{1}{\frac{4}{3}\pi R_d^3} = -3D_0 \frac{1}{R_d} \frac{\rho_{v,*} - \rho_v(R_d)}{R_E - R_d}. \tag{12}$$

Definition (12) represents a possible choice for the relaxation rate, but should ideally be reviewed with the help of measurements.

Altogether, substituting equations (7), (10) and (12) into (6) yields

$$\frac{\mathrm{d}\rho_{v,*}}{\mathrm{d}t} = \frac{\mathrm{d}}{\mathrm{d}t}\left(\rho_v(R_d)\right) + N_d \frac{J_d}{ZV} - \frac{3D_0}{R_d}\frac{\rho_{v,*} - \rho_v(R_d)}{R_E - R_d} = \frac{\mathrm{d}}{\mathrm{d}t}\left(\rho_v(R_d)\right) - \frac{N_d}{\alpha_d ZV}\frac{\mathrm{d}m_d}{\mathrm{d}t} - \frac{3D_0}{R_d}\frac{\rho_{v,*} - \rho_v(R_d)}{R_E - R_d} \tag{13}$$

for the rate of change of coupling water vapor density.

To define the volume $V$ in equation (10), we first give an alternate interpretation of the rate $\frac{J_d}{ZV}$. Let

$$V_S := \frac{4}{3}\pi\left((R_d + r_E)^3 - (R_d - r_E)^3\right) \tag{14}$$

denote the volume of the artificial spherical shell, obtained by "bloating" the sphere of radius $R_d$ about the ice crystal, see figure 3. Using equation (9), the rate may be rewritten as

$$\frac{J_d}{ZV} = \frac{J_d}{V_S}\frac{\frac{4}{3}\pi r_E^3}{V}. \tag{15}$$

The first factor represents the rate if all exchanged water vapor would modify the vapor density inside the artificial spherical shell. The second factor amounts to a scaling of the first factor. Since the volume $V$ is the volume, where the released water vapor diffuses in unit time, we define it as a scaled influence sphere about the droplet as

$$V := \frac{4}{3}\pi(nr_E)^3 = \frac{4}{3}\pi n^3 r_E^3 \tag{16}$$



with the scaled influence radius $nr_E$, leading to the rate $\frac{J_d}{ZV} = \frac{J_d}{V_S}\frac{1}{n^3}$. With this definition, only a fraction $\frac{1}{n^3}$ of the released water vapor from the droplet will affect the coupling value inside the artificial spherical shell. The choice of the parameter $n$ will be discussed in subsection 2.2.4 below.

For the radii of influence $R_E, r_E$ we define

$$R_E := R_i + \mathcal{D}_i, \tag{17a}$$

$$r_E := r_d + \mathcal{D}_d \tag{17b}$$

for some positive constants $\mathcal{D}_i, \mathcal{D}_d$. Likewise, we define the coupling radius by

$$R_d := R_i + \mathcal{D}_0 \tag{18}$$

with $0 < \mathcal{D}_0 < \mathcal{D}_i$. With these definitions we can state the complete ODE system as

$$\frac{\mathrm{d}m_d}{\mathrm{d}t} = 4\pi\alpha_d D_0 r_d r_E \frac{\rho_{v,*} - \rho_{v,d}}{r_E - r_d}, \tag{19a}$$

$$\frac{\mathrm{d}T_d}{\mathrm{d}t} = \frac{4\pi r_d r_E}{m_d c_{p,l}(T_d)(r_E - r_d)}\Big(\alpha_d D_0 L_{lv}(T_d)(\rho_{v,*} - \rho_{v,d}) + K_0(T_* - T_d)\Big), \tag{19b}$$

$$\frac{\mathrm{d}M_i}{\mathrm{d}t} = 4\pi\alpha_i D_0 R_i R_d \frac{\rho_{v,*} - \rho_{v,i}}{R_d - R_i}, \tag{19c}$$

$$\frac{\mathrm{d}T_i}{\mathrm{d}t} = \frac{4\pi R_i R_d}{M_i c_{p,i}(T_i)(R_d - R_i)}\Big(\alpha_i D_0 L_{iv}(T_i)(\rho_{v,*} - \rho_{v,i}) + K_0(T_* - T_i)\Big), \tag{19d}$$

$$\frac{\mathrm{d}\rho_{v,*}}{\mathrm{d}t} = \frac{\mathrm{d}}{\mathrm{d}t}\Big(\rho_v(R_d)\Big) + N_d \frac{J_d}{ZV} - \frac{3D_0}{R_d}\frac{\rho_{v,*} - \rho_v(R_d)}{R_E - R_d}, \tag{19e}$$

$$\frac{\mathrm{d}T_*}{\mathrm{d}t} = \frac{\mathrm{d}}{\mathrm{d}t}\Big(T(R_d)\Big). \tag{19f}$$

Substituting $R_d$ from (18) into the field representations in equation (1) yields an expression of $\rho_v(R_d)$ and $T(R_d)$, allowing to compute the required derivatives. Keeping equations (17) and (18) in mind, we arrive at

$$\frac{\mathrm{d}}{\mathrm{d}t}\Big(\rho_v(R_d)\Big) = \frac{R_E}{R_E - R_i}\left(1 - \frac{R_i}{R_d}\right)\frac{\mathrm{d}\rho_{v,\infty}}{\mathrm{d}t} + \frac{R_i}{R_E - R_i}\left(\frac{R_E}{R_d} - 1\right)\frac{\mathrm{d}\rho_{v,i}}{\mathrm{d}t} + \frac{\rho_{v,i} - \rho_{v,\infty}}{R_E - R_i}\left(\frac{R_E + R_i}{R_d} - \frac{R_i R_E}{R_d^2} - 1\right)\frac{\mathrm{d}R_i}{\mathrm{d}t} \tag{20}$$

and

$$\frac{\mathrm{d}}{\mathrm{d}t}\Big(T(R_d)\Big) = \frac{R_E}{R_E - R_i}\left(1 - \frac{R_i}{R_d}\right)\frac{\mathrm{d}T_\infty}{\mathrm{d}t} + \frac{R_i}{R_E - R_i}\left(\frac{R_E}{R_d} - 1\right)\frac{\mathrm{d}T_i}{\mathrm{d}t} + \frac{T_i - T_\infty}{R_E - R_i}\left(\frac{R_E + R_i}{R_d} - \frac{R_i R_E}{R_d^2} - 1\right)\frac{\mathrm{d}R_i}{\mathrm{d}t}. \tag{21}$$

The derivative of the ice crystal radius may be calculated using the equation for the mass of a spherical ice crystal $M_i = \frac{4}{3}\pi\rho_i(T_i)R_i^3$ with ice density $\rho_i$, yielding

$$\frac{\mathrm{d}R_i}{\mathrm{d}t} = \frac{1}{4\pi\rho_i(T_i)R_i^2}\frac{\mathrm{d}M_i}{\mathrm{d}t} = \frac{\alpha_i D_0}{\rho_i(T_i)}\frac{R_d}{R_i}\frac{\rho_{v,*} - \rho_{v,i}}{R_d - R_i} \tag{22}$$





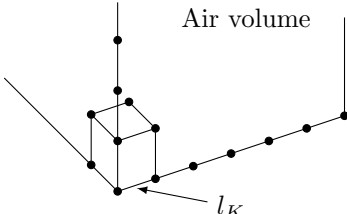

**Figure 4.** Schematic of a perfect spatially homogeneous droplet distribution within an air parcel, where the dots indicate the positions of the droplets.

where we neglected the derivative of the ice density $\rho_i$.

Neglecting chemical composition and curvature effects of the ice crystal, we approximate the saturation vapor density $\rho_{v,i}$ at the surface by

$$\rho_{v,i} \approx \frac{p_{i,\infty}(T_i)}{R_v T_i}. \tag{23}$$

Using the Clausius-Clapeyron equation (Lamb and Verlinde, 2011, equations (4.35) and (4.36)), we obtain the temporal derivative

$$\frac{d\rho_{v,i}}{dt} \approx \frac{d}{dt}\left(\frac{p_{i,\infty}(T_i)}{R_v T_i}\right)\frac{dT_i}{dt} = \frac{p_{i,\infty}(T_i)}{R_v T_i^2}\left(\frac{L_{iv}(T_i)}{R_v T_i} - 1\right)\frac{dT_i}{dt}. \tag{24}$$

## 2.2 Choice of Parameters

In this subsection we discuss a possible choice of the parameters $\mathcal{D}_i$, $\mathcal{D}_d$, $\mathcal{D}_0$, $N_d$ and $n$ from equations (17), (18), (19e), (16), respectively. We estimate possible values for these parameters using typical environmental conditions in a mixed-phase cloud with ambient temperature $T_\infty = -15\,^\circ\mathrm{C}$, ambient pressure $p_\infty = 650\,\mathrm{hPa}$, ambient saturation ratio $S_\infty = 1.01$, ice crystal radius $R_i = 100\,\mu\mathrm{m}$ and droplet radius $r_d = 10\,\mu\mathrm{m}$.

### 2.2.1 Parameter for Droplet Distance

Parameter $\mathcal{D}_0$ in equation (18) defines the distance from the droplet center to the ice crystal surface. To estimate this parameter, we assume a perfect spatially homogeneous distribution of $\mathcal{N} = 1000\,\mathrm{cm}^{-3}$ droplets inside the cloud volume. This droplet number is higher than typical observed values, but since our focus is on local droplet accumulations around ice crystals, we use this higher value. A perfect spatially homogeneous distribution of droplets is illustrated in figure 4, where all droplets are on the vertices of small cuboids with edge length

$$l_K := \frac{1}{\sqrt[3]{\mathcal{N}} - 1}. \tag{25}$$

If the ice crystal is located in the center of such a cuboid, the distance from the midpoint of the ice crystal to the midpoint of the nearest droplet is given by $\frac{\sqrt{3}}{2}l_K$, yielding an estimate for $R_d = R_i + \mathcal{D}_0$. Substituting values, we arrive at $\mathcal{D}_0 \approx 7.67 \cdot 10^{-4}\,\mathrm{m} = 767\,\mu\mathrm{m}$ being approximately 7.5 times ice radii, so $\mathcal{D}_0 = 7.5 R_i$.




### 2.2.2 Parameter for Radii of the Influence Spheres

To estimate the distance parameter $\mathcal{D}_i$, determining the radius of the influence sphere of the ice crystal, we use the representation

$$\rho_v(R) = \rho_{v,\infty} - \frac{R_i(\rho_{v,\infty} - \rho_{v,i})}{R} \tag{26}$$

of the water vapor field obtained from Maxwellian growth theory. Let $\xi > 0$ denote a chosen maximal relative deviation of $\rho_v(R)$ from the environmental value $\rho_{v,\infty}$, we seek the radius $R_E$ such that

$$\frac{|\rho_v(R) - \rho_{v,\infty}|}{\rho_{v,\infty}} \leq \xi \tag{27}$$

is satisfied for $R \geq R_E$. Substituting (26) into (27) yields the condition

$$\frac{R_i}{\xi \rho_{v,\infty}} |\rho_{v,\infty} - \rho_{v,i}| \leq R_E. \tag{28}$$

Neglecting effects of chemical substances and curvature, we estimate the surface water vapor density as $\rho_{v,i} = \frac{p_{i,\infty}(T_\infty)}{R_v T_\infty}$.

Using a maximal relative deviation $\xi = 10^{-3}$, we arrive at $\mathcal{D}_i \approx 0.0144\,\text{m}$ being approximately 144 ice radii, so $\mathcal{D}_i = 144 R_i$. Using the same approach for the droplets, we obtain $\mathcal{D}_d \approx 8.9 \cdot 10^{-5}\,\text{m}$ being approximately 9 droplet radii, so $\mathcal{D}_d = 9 r_d$. We chose $\xi = 10^{-3}$ as the maximal relative deviation, because the relative deviations

$$\frac{|\rho_{v,\infty} - \rho_{v,i}|}{\rho_{v,\infty}} \quad \text{and} \quad \frac{|\rho_{v,\infty} - \rho_{v,d}|}{\rho_{v,\infty}} \tag{29}$$

are already of the order $1.4 \cdot 10^{-1}$ and $10^{-2}$, respectively, for the conditions stated at the beginning of the subsection. For the case of a droplet, Reiss and La Mer (1950) and Reiss (1951) suggest a value of ten times droplet diameter for the radius of influence. With our choice of the relative deviation, we obtain the same order of magnitude.

### 2.2.3 Number Parameter

In order to estimate a typical number $N_d$ of droplets within the influence sphere about the ice crystal, we employ the typical
droplet number concentration $\mathcal{N} = 70\,\text{cm}^{-3}$ in mixed-phase clouds (Korolev et al., 2003; Zhao and Lei, 2014). Using $\mathcal{D}_i = 144 R_i$, we obtain a droplet number

$$N_d = \mathcal{N} \frac{4}{3} \pi (R_i + \mathcal{D}_i)^3 = \mathcal{N} \frac{4}{3} \pi (145 \cdot R_i)^3 \approx 894. \tag{30}$$

Not all of these droplets are precisely at interaction distance $R_d$ from the midpoint of the ice crystal and the influence of droplets decreases with increasing distance from ice crystal center. Qualitatively, only the droplets closest to the ice crystal influence its
growth behavior significantly, see Baumgartner and Spichtinger (2017b, their figure 6). We employ the conservative assumption that all droplets with distances smaller than $\frac{R_d}{3}$ to have larger influence on the ice crystal and consequently define the droplet number $N_d$ as

$$N_d = \mathcal{N} \frac{4}{3} \pi \left( \frac{R_i + \mathcal{D}_0}{3} \right)^3 = \mathcal{N} \frac{4}{3} \pi \left( \frac{145}{3} R_i \right)^3 \approx 32, \tag{31}$$





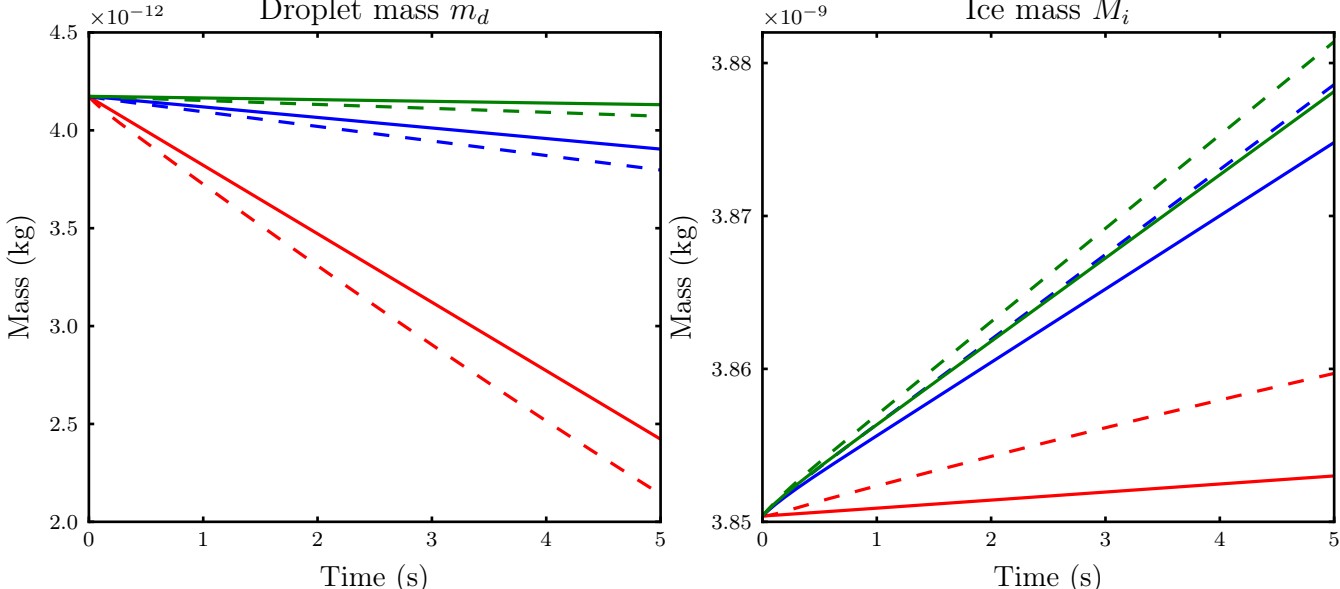

**Figure 5.** Comparison of the temporal evolution of hydrometeor masses obtained with the new model from equation (19) (dashed) and the reference model (solid) for the case of $N_d = 14$ droplets per influence sphere. Ambient saturation ratios are $S_\infty = 0.86$ (red), $S_\infty = 0.99$ (blue) and $S_\infty = 1.01$ (green).

motivating the choice $N_d = 40$. We remark that this droplet number is highly variable in real clouds due to turbulent effects. If one knew an expression, describing the change of droplet number inside the influence sphere, one could also use a variable number instead of a constant.

### 2.2.4 Distribution Parameter

Parameter $n$ in equation (16) is a critical parameter of the whole model, since it determines directly the strength of the interaction between the ice crystal and the droplet. According to the interpretation given above, it describes which fraction of released water vapor from an evaporating droplet is incorporated into the artificial spherical shell around the ice crystal and consequently influences the growth of the ice crystal, see equation (15). If $n = 1$, all released water vapor is incorporated in the artificial spherical shell, if $n > 1$ only a fraction of $\frac{1}{n^3}$. All remaining water vapor is released to the environment. In this

study, we use a value $n = 1.8$ and justify this choice with the help of direct simulations using the reference model described in Baumgartner and Spichtinger (2017b), since the authors are not aware of any direct measurements.

    We compare the temporal evolution of the ice crystal and droplet mass obtained from the reference model with results of the model presented in section 2.1 using the two droplet numbers $N_d = 14$ and $N_d = 38$. For calculations with the reference model, we distributed the droplets uniformly along a sphere with radius $7.5R_i$ about the ice crystal. Figure 5 shows the temporal

evolution of droplet and ice crystal mass at ambient saturation ratios $S_\infty = 0.86$, $S_\infty = 0.99$ and $S_\infty = 1.01$ for the case of





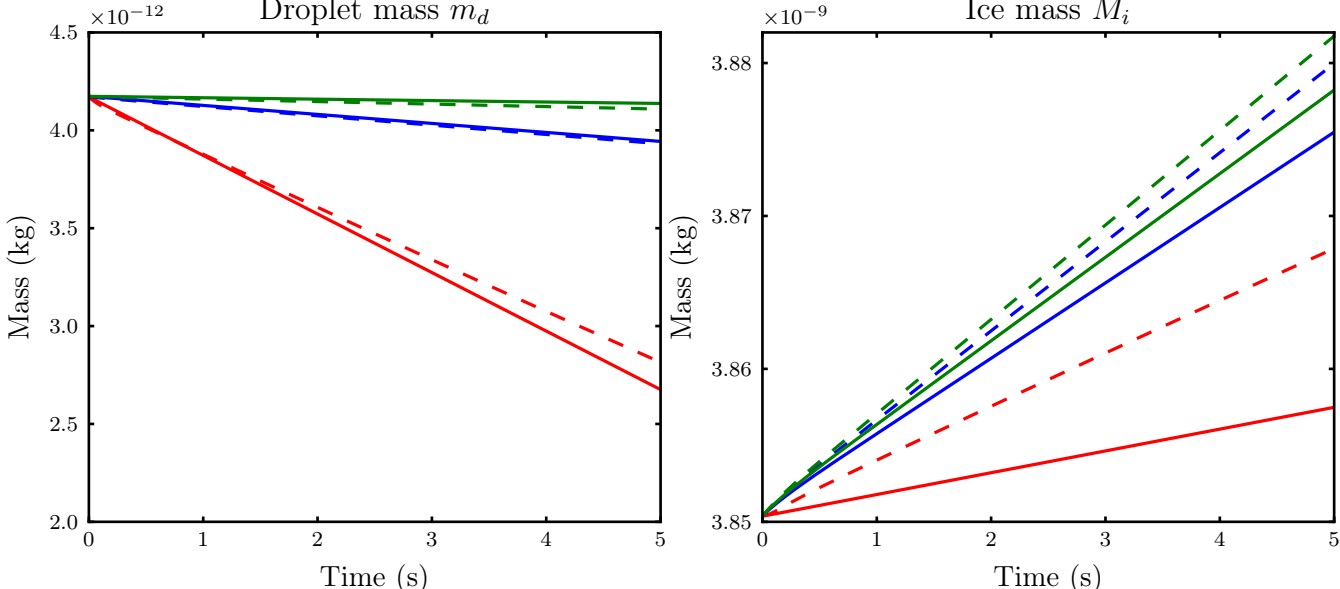

**Figure 6.** As in figure 5, but for the case of $N_d = 38$ droplets per influence sphere.

$N_d = 14$ droplets inside the influence sphere of the ice crystal. For the newly presented model, we used the parameter value $n = 1.8$. The curves agree well although the droplet masses are slightly underestimated and the ice crystal mass is slightly overestimated. Changing the number of droplets to $N_d = 38$, yields similar results, shown in figure 6. Numerical experiments reveal, by choosing other values for $n$, we can achieve better agreement of either the ice mass or droplet mass curves in figures 5 and 6, but not for both simultaneously. Moreover, our choice of parameter shows also satisfactory results in other mass and humidity regimes.

### 2.3 Simplifications

The model in equation (19) consists of many equations, especially it keeps track of the temperature of the individual hydrometeors. In Maxwellian growth theory, the equation for the temperature of a hydrometeor is eliminated. We can simplify the model equations (19) in a similar way. Proceeding as in the derivation of the Maxwellian growth equations, we can eliminate the temperature equations of the hydrometeors. The modified equations for the hydrometeor masses are then given by

$$\frac{dm_d}{dt} = \frac{4\pi}{\left(\frac{L_{lv}(T_\infty)}{R_v T_\infty} - 1\right)\frac{L_{lv}(T_\infty)}{K_0 T_\infty} + \frac{R_v T_\infty}{\alpha_d D_0 p_{l,\infty}(T_\infty)}} \frac{r_d r_E}{r_E - r_d}(S_* - 1) =: G_l \frac{r_d r_E}{r_E - r_d}(S_* - 1) \tag{32}$$

for droplet mass and

$$\frac{dM_i}{dt} = \frac{4\pi}{\left(\frac{L_{iv}(T_\infty)}{R_v T_\infty} - 1\right)\frac{L_{iv}(T_\infty)}{K_0 T_\infty} + \frac{R_v T_\infty}{\alpha_i D_0 p_{i,\infty}(T_\infty)}} \frac{R_i R_d}{R_d - R_i}(S_{*,i} - 1) =: G_i \frac{R_i R_d}{R_d - R_i}(S_{*,i} - 1) \tag{33}$$





for ice crystal mass with the saturation ratios $S_* = \frac{\rho_{v,*} R_v T_\infty}{p_{l,\infty}(T_\infty)}$ and $S_{*,i} = \frac{\rho_{v,*} R_v T_\infty}{p_{i,\infty}(T_\infty)}$, respectively. In addition, we neglect the temperature equation (19f) for the coupling temperature $T_*$ in the governing system (19). Since $T_*$ also appears in the calculation of the unperturbed value $\rho_v(R_d)$ through the equilibrium vapor density at the ice crystal surface, we have to modify its calculation from equation (1). We simplify the vapor density field from equation (1) by considering the case $R_E \to \infty$, representing the observation $R_i \ll R_E$. The vapor density at the ice crystal surface is approximated as in (23), yielding

$$\rho_v(R_d) \approx \rho_{v,\infty} + (\rho_{v,i} - \rho_{v,\infty}) \frac{R_i}{R_d} \approx \rho_{v,\infty} + \left( \frac{p_{i,\infty}(T_\infty)}{R_v T_\infty} - \rho_{v,\infty} \right) \frac{R_i}{R_d}. \tag{34}$$

Using the Clausius-Clapeyron equation and (22), we compute the required time derivative of (34) as

$$\begin{aligned}
\frac{\mathrm{d}}{\mathrm{d}t}\left( \rho_v(R_d) \right) &\approx \left( 1 - \frac{R_i}{R_d} \right) \frac{\mathrm{d}\rho_{v,\infty}}{\mathrm{d}t} + \frac{R_i}{R_d} \frac{\mathrm{d}}{\mathrm{d}t}\left( \frac{p_{i,\infty}(T_\infty)}{R_v T_\infty} \right) + \left( \frac{p_{i,\infty}(T_\infty)}{R_v T_\infty} - \rho_{v,\infty} \right) \frac{R_d - R_i}{R_d^2} \frac{\mathrm{d}R_i}{\mathrm{d}t} \\
&= \left( 1 - \frac{R_i}{R_d} \right) \frac{\mathrm{d}\rho_{v,\infty}}{\mathrm{d}t} + \frac{R_i}{R_d} \frac{p_{i,\infty}(T_\infty)}{R_v T_\infty^2} \left( \frac{L_{iv}(T_\infty)}{R_v T_\infty} - 1 \right) \frac{\mathrm{d}T_\infty}{\mathrm{d}t} + \left( \frac{p_{i,\infty}(T_\infty)}{R_v T_\infty} - \rho_{v,\infty} \right) \frac{R_d - R_i}{4\pi \rho_i(T_\infty) R_i^2 R_d^2} \frac{\mathrm{d}M_i}{\mathrm{d}t}.
\end{aligned} \tag{35}$$

Accepting those further approximations, we arrive at the simplified system

$$\frac{\mathrm{d}m_d}{\mathrm{d}t} = G_l \frac{r_d r_E}{r_E - r_d} (S_* - 1), \tag{36a}$$

$$\frac{\mathrm{d}M_i}{\mathrm{d}t} = G_i \frac{R_i R_d}{R_d - R_i} (S_{*,i} - 1), \tag{36b}$$

$$\frac{\mathrm{d}\rho_{v,*}}{\mathrm{d}t} = \frac{\mathrm{d}}{\mathrm{d}t}\left( \rho_v(R_d) \right) - \frac{N_d}{\alpha_d Z V} \frac{\mathrm{d}m_d}{\mathrm{d}t} - \frac{3 D_0}{R_d} \frac{\rho_{v,*} - \rho_v(R_d)}{R_E - R_d}, \tag{36c}$$

where only the three prognostic variables $m_d$, $M_i$ and $\rho_{v,*}$ are left. For a bulk model, we would add just one additional equation.

## 2.4 Discussion of the Model Ansatz

Already Srivastava (1989) criticized the use of Maxwellian growth theory for the description of droplet growth by diffusion. He advocated the use of local quantities instead of the environmental conditions, since water vapor density is spatially variable and the droplet growth heavily depends on its precise value. By using local variables, one can take those variations into account and compute growth rates of hydrometeors more accurately. He connected the local value for vapor density to the environmental value through a relaxation of local conditions to the environmental conditions, similarly as we did in equation (12) for $I_3$. Srivastava proposed to consider local volumes about every hydrometeor and to compute its individual growth rate by using the local vapor density. We adopted this approach in our model by introducing the "spheres of influence". Similar ideas were employed in Marshall and Langleben (1954), were the authors also consider a local volume about an ice crystal. In contrast to our formulation, they assume a continuous droplet distribution inside the local volume. This method avoids additional growth equations for the nearby droplets. Our approach combines ideas of both former studies, where we focused on a model formulation which only incorporates values that are typically known in a numerical cloud model.





In subsection 2.2, we estimated possible values for the parameters of our model. To our best knowledge, there are no direct measurements of the local interactions of a single ice crystal with surrounding droplets available, so we cannot compare our parameter choice with real measurements. Instead we can consider two separate extreme cases, since equation (19e) for the coupling water vapor density has the generic representation

$$\frac{\mathrm{d}\rho_{v,*}}{\mathrm{d}t} = \frac{\mathrm{d}}{\mathrm{d}t}\left(\rho_v(R_d)\right) - \frac{N_d}{ZV}\frac{\mathrm{d}m_d}{\mathrm{d}t} - C\left(\rho_{v,*} - \rho_v(R_d)\right) \tag{37}$$

for a constant $C > 0$. In our model derivation, we set $C := \frac{3D_0}{R_d(R_E - R_d)}$. The two extreme cases are constructed by choosing $C = 0$ or $C \to \infty$. In the first case $C = 0$, relaxation of the coupling value $\rho_{v,*}$ to the equilibrium value $\rho_v(R_d)$ is neglected and therefore $\rho_{v,*}$ is solely changed by the growth or evaporation of the hydrometeors. The second case $C \to \infty$ corresponds to an instantaneous relaxation to equilibrium. This is basically the same behavior as in the classical treatment using Maxwellian growth theory. We emphasize that both extreme cases are rather nonphysical, but they may serve to assess the possible strength of the local interactions. However, we believe that ansatz (37) is able to capture the essential behavior, possibly after another correction of the coefficient $\frac{1}{ZV}$ with the help of measurements.

## 3   Incorporation into a Parcel Model

In this section, we incorporate the new model of section 2 into a parcel model. Let $p_\infty$ and $T_\infty$ denote pressure and temperature of the air parcel. We divide the total water mass into mass of water vapor $M_v$, mass of liquid water $M_l$ and mass of ice $M_{\mathrm{ice}}$. In addition, the air parcel contains a mass $M_a$ of dry air. Since the air parcel is assumed as thermodynamically closed, the mass $M_a$ of dry air and the total water mass $M_v + M_l + M_{\mathrm{ice}}$ are constant. Instead of the masses, we consider the mixing-ratios $q_x := \frac{M_x}{M_a}$ for $x \in \{v, l, \mathrm{ice}\}$.

### 3.1   Description of the Parcel Model

Variations in pressure $p_\infty$ are governed by the equation

$$\frac{\mathrm{d}p_\infty}{\mathrm{d}t} = \frac{\mathrm{d}p_\infty}{\mathrm{d}z}\frac{\mathrm{d}z}{\mathrm{d}t} = -g\frac{p_\infty}{\overline{R}T_\infty}w, \tag{38}$$

obtained by applying the equation for hydrostatic equilibrium. Coordinate $z$ denotes the height, $g$ the acceleration of gravity, $w$ the vertical velocity and $\overline{R}$ the gas constant for moist air, given by

$$\overline{R} := R_a\left(1 + \frac{1 - \varepsilon}{\varepsilon}\frac{q_v}{1 + q_v}\right) \tag{39}$$

where $\varepsilon := \frac{R_a}{R_v}$ and $R_a, R_v$ denote the gas constants for dry air and water vapor, respectively. A change in the air parcel temperature $T_\infty$ has two contributions. The first contribution comes from the adiabatic vertical motion of the air parcel and is given by (for example Wang, 2013, chapter 12)

$$\left.\frac{\mathrm{d}T_\infty}{\mathrm{d}t}\right|_{\mathrm{adiabatic}} = -\frac{g}{c_p}w \tag{40}$$





where the specific heat capacity of moist air is given by (Rogers and Yau, 1989, chapter 2)

$$\overline{c_p} := c_{p,a} \left( 1 + \left( \frac{c_{p,v}}{c_{p,a}} - 1 \right) \frac{q_v}{1 + q_v} \right) \tag{41}$$

and $c_{p,a}$, $c_{p,v}$ denote the specific heat capacity of dry and moist air, respectively. If condensation of water vapor takes place, we have to include latent heat effects. Let $\omega_k$ be a hydrometeor, then its temperature $T_k$ changes as

$$5 \quad c_{p,k} m_k \frac{\mathrm{d}T_k}{\mathrm{d}t} = L \frac{\mathrm{d}m_k}{\mathrm{d}t} + \int_{\partial \omega_k} K \nabla T \cdot N \, \mathrm{d}\sigma, \tag{42}$$

where $m_k$ is the mass of hydrometeor $\omega_k$, $L$ the latent heat of a phase change, $K$ the thermal conductivity of air and $N$ the outer normal to the surface $\partial \omega_k$ of the hydrometeor. The surface integral in (42) accounts for heat conduction from the hydrometeor to the air parcel. The amount of heat $-\mathrm{d}Q_{\text{conduction}}$, delivered from the single hydrometeor $\omega_k$ to the air parcel, is given by the rate

$$10 \quad -\frac{\mathrm{d}Q_{\text{conduction}}}{\mathrm{d}t} = \int_{\partial \omega_k} K \nabla T \cdot N \, \mathrm{d}\sigma. \tag{43}$$

This amount of heat changes the temperature $T_\infty$ of the air parcel according to $\overline{c_p} \left( M_v + M_a \right) \mathrm{d}T_\infty = -\mathrm{d}Q_{\text{conduction}}$. Inserting (43) and summing over all hydrometeors yields the rate

$$\left. \frac{\mathrm{d}T_\infty}{\mathrm{d}t} \right|_{\text{latent}} = -\frac{1}{\overline{c_p} \left( M_v + M_a \right)} \sum_k \int_{\partial \omega_k} K \nabla T \cdot N \, \mathrm{d}\sigma \tag{44}$$

of latent heating. Combining both contributions (40) and (44) yields the final rate of the temperature change

$$15 \quad \frac{\mathrm{d}T_\infty}{\mathrm{d}t} = \left. \frac{\mathrm{d}T_\infty}{\mathrm{d}t} \right|_{\text{adiabatic}} + \left. \frac{\mathrm{d}T_\infty}{\mathrm{d}t} \right|_{\text{latent}} = -\frac{g}{c_p} w - \frac{1}{\overline{c_p} (M_v + M_a)} \sum_k \int_{\partial \omega_k} K \nabla T \cdot N \, \mathrm{d}\sigma \tag{45}$$

where the sum expands over all hydrometeors. In the literature, one finds a slightly different equation, where the surface integral is replaced by $-L \frac{\mathrm{d}m_k}{\mathrm{d}t}$ (for example Pruppacher and Klett, 1997, equation 12.15). Assuming all hydrometeors to have reached their equilibrium temperature as is done in classical Maxwellian growth theory, the time derivative on the left hand side of (42) vanishes and equation (45) reduces to the equation from literature.

20      We further divide the liquid water mass $M_l$ into the mass $M_l^i$ of all droplets located in an influence sphere of some ice crystal and the mass $M_l^o$ of all droplets outside of every ice crystal influence sphere. From now on, we assume monodisperse mass distributions for the ice crystals with number concentration $\mathcal{N}_{\text{ice}}$, for the droplets inside the ice crystal influence spheres with number concentration $\mathcal{N}_d^i$ and for the droplets outside of every ice crystal influence sphere with number concentration $\mathcal{N}_d^o$. As for the liquid water mass, we divide the corresponding droplet temperatures in $T_d^i$, $T_d^o$ and the liquid water mixing-ratios in

25   $q_l^i := \frac{M_l^i}{M_a}$ and $q_l^o := \frac{M_l^o}{M_a}$. With these definitions, we obtain the relations

$$q_{\text{ice}} = \mathcal{N}_{\text{ice}} M_{\text{ice}}, \quad \text{and} \quad q_l = q_l^o + q_l^i = \mathcal{N}_d^o m_d^o + N_d \mathcal{N}_{\text{ice}} m_d^i \tag{46}$$

where $N_d$ is the number of droplets inside the influence sphere of an ice crystal.



Using this notation and the assumption of spherical droplets, we can evaluate the surface integrals in (45) to obtain

$$
\begin{aligned}
\frac{\mathrm{d}T_\infty}{\mathrm{d}t} = &-\frac{g}{c_p}w - \frac{1}{\overline{c_p}(M_v + M_a)}\Big(\mathcal{N}_d^o M_a 4\pi r_d^o K_0 (T_\infty - T_d^o) \\
&+ N_d \mathcal{N}_\mathrm{ice} M_a 4\pi r_d^i K_0 (T_\infty - T_d^i) + \mathcal{N}_\mathrm{ice} M_a 4\pi R_i K_0 (T_\infty - T_i)\Big) \\
= &-\frac{g}{c_p}w - \frac{4\pi K_0 M_a}{\overline{c_p}(M_v + M_a)}\Big(\mathcal{N}_d^o r_d^o (T_\infty - T_d^o) \\
&+ N_d \mathcal{N}_\mathrm{ice} r_d^i (T_\infty - T_d^i) + \mathcal{N}_\mathrm{ice} R_i (T_\infty - T_i)\Big) \\
= &-\frac{g}{c_p}w - \frac{4\pi K_0}{\overline{c_p}(1 + q_v)}\Big(\mathcal{N}_d^o r_d^o (T_\infty - T_d^o) \\
&+ N_d \mathcal{N}_\mathrm{ice} r_d^i (T_\infty - T_d^i) + \mathcal{N}_\mathrm{ice} R_i (T_\infty - T_i)\Big).
\end{aligned}
\tag{47}
$$

The mixing-ratio for water vapor is determined by the conservation of mass and reads

$$
\frac{\mathrm{d}q_v}{\mathrm{d}t} = -\frac{\mathrm{d}q_l^o}{\mathrm{d}t} - \frac{\mathrm{d}q_l^i}{\mathrm{d}t} - \frac{\mathrm{d}q_\mathrm{ice}}{\mathrm{d}t} = -\mathcal{N}_d^o \frac{\mathrm{d}m_d^o}{\mathrm{d}t} - \mathcal{N}_\mathrm{ice}\left(N_d \frac{\mathrm{d}m_d^i}{\mathrm{d}t} + \frac{\mathrm{d}M_\mathrm{ice}}{\mathrm{d}t}\right).
\tag{48}
$$

5   The equations for the other mixing-ratios are given by

$$
\frac{\mathrm{d}q_\mathrm{ice}}{\mathrm{d}t} = \mathcal{N}_\mathrm{ice}\frac{\mathrm{d}M_\mathrm{ice}}{\mathrm{d}t},
\tag{49a}
$$

$$
\frac{\mathrm{d}q_l^o}{\mathrm{d}t} = \mathcal{N}_d^o \frac{\mathrm{d}m_d^o}{\mathrm{d}t},
\tag{49b}
$$

$$
\frac{\mathrm{d}q_l^i}{\mathrm{d}t} = N_d \mathcal{N}_\mathrm{ice}\frac{\mathrm{d}m_d^i}{\mathrm{d}t}.
\tag{49c}
$$

503.61377pt

## 10  3.2  Results

Using the parcel model we carry out several simulations, where we vary the following (initial) parameters:

- Coupling distance $R_d$ through parameter

$$
\mathcal{D}_0 \in \{5R_i,\ 15R_i,\ 30R_i,\ 50R_i,\ 100R_i\}
\tag{50}
$$

with initial ice crystal radius $R_i = 100\,\mu\mathrm{m}$.

15    – Number of droplets inside the influence sphere of the ice crystals

$$
N_d \in \{40,\ 100,\ 200,\ 500\}.
\tag{51}
$$

- Ambient saturation ratio

$$
S_\infty \in \{0.847,\ 0.932,\ 1.01\}
\tag{52}
$$

with respect to water. These values correspond to the ambient ice saturation ratios

20    $$
S_{\infty,i} \in \{0.98,\ 1.079,\ 1.169\}.
\tag{53}
$$





- Vertical velocity

$$w \in \left\{ 0\,\mathrm{ms}^{-1}, -1\,\mathrm{ms}^{-1}, 1\,\mathrm{ms}^{-1} \right\}. \tag{54}$$

The initial values for ambient temperature and pressure were $T_\infty = -15\,^\circ\mathrm{C}$ and $p_\infty = 650\,\mathrm{hPa}$, respectively, resembling typical environmental conditions for a mixed-phase cloud. Reliable values for the droplet number $N_d$ inside every influence sphere of the ice crystals are difficult to estimate, since this depends heavily on small-scale turbulence. Therefore, any of the estimated values in subsection 2.2.3 between $N_d = 30$ and $N_d = 800$ is possible, explaining the choices in (51).

Characteristics of mixed-phase clouds are, among others, reported in Fleishauer et al. (2002); Hobbs et al. (2001); Pinto et al. (2001); Noh et al. (2013); Zhao and Lei (2014); Lloyd et al. (2015); Verlinde et al. (2007). The study by Korolev et al. (2003) collects measurement data from various research flights. In all studies, we observe a large scattering of the microphysical parameters, especially in liquid water content (LWC) and ice water content (IWC). We use the typical values $\mathrm{LWC} = 0.045\,\mathrm{gm}^{-3}$ (Korolev et al., 2003) and $\mathrm{IWC} = 0.013\,\mathrm{gm}^{-3}$ (Fleishauer et al., 2002). A typical droplet radius is given by $10\,\mu\mathrm{m}$. Variability in size of the ice crystals is much larger, but on average, pristine ice crystals in mixed-phase clouds tend to be smaller than in ice clouds (Korolev et al., 2003) and we use again a value of $100\,\mu\mathrm{m}$ as initial radius.

In the sequel, we present simulation results ordered by vertical velocity. All figures contain three curves: the red curve represents the solution of the new system (19), the cyan and blue curve represent the solutions of the same system, where the equation for the coupling water vapor density (19e) is replaced by (37) with $C = 0$ and $C \to \infty$, respectively. Consequently, those two curves correspond to the extreme cases without local relaxation to equilibrium (case $C = 0$) and instantaneous relaxation to equilibrium (case $C \to \infty$). The spreading between the two curves show the spectrum of possible values for different choices of the parameter $C$.

### 3.2.1 Vertical Velocity $w = 0\,\mathrm{ms}^{-1}$

In Baumgartner and Spichtinger (2017b), the authors documented the largest effect of surrounding droplets on ice growth in an ice subsaturated environment, because the evaporating droplets can deliver enough water vapor towards the ice crystal to produce a local supersaturation with respect to ice, allowing the crystal to grow instead of evaporate. With the new model, we also observe a similar behavior. Consider for example the case of $N_d = 40$ droplets per influence sphere of any ice crystal with the small droplet distance $\mathcal{D}_0 = 5R_i = 5 \cdot 100\,\mu\mathrm{m}$ from the ice crystal and ambient saturation ratio $S_\infty = 0.847$, being subsaturated with respect to ice and water. Figure 7 shows the temporal evolution of ice mixing-ratio $q_i$ and liquid water mixing-ratio $q_l$. We observe an increasing ice mixing-ratio $q_i$, showing the aforementioned local interaction. As long as not all droplets inside the influence spheres are evaporated, the red curve for ice mixing-ratio in the left panel of figure 7 coincides with the cyan curve, indicating the extreme case without local relaxation. This means the evaporating droplets inside the influence sphere of the ice crystal provide enough water vapor to mostly compensate diffusion of water vapor to the environment. The red curve in the right panel of figure 7 shows the temporal evolution of liquid water mixing-ratio $q_l$. At the first kink of this curve, all droplets outside of the influence spheres are evaporated ($q_l^o = 0$) and at the second kink also the droplets inside the influence spheres are evaporated, indicating $q_l = q_l^o = q_l^i = 0$. Although the environment is subsaturated with respect to ice,



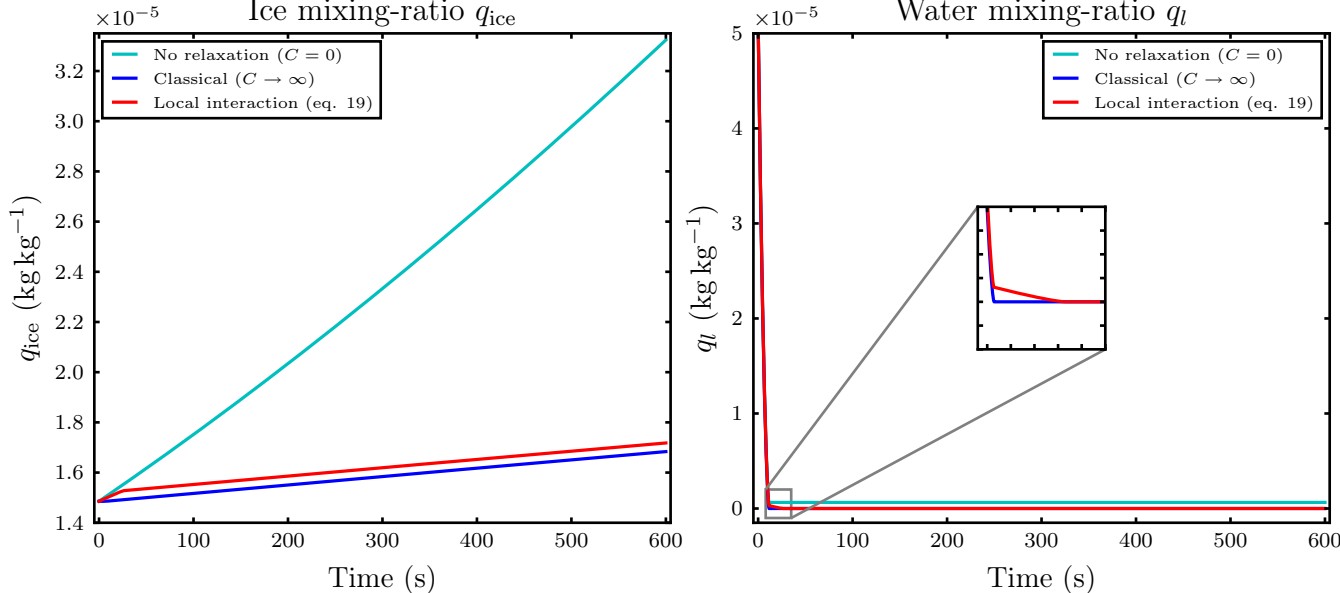

**Figure 7.** Parcel model simulation with $w = 0\,\mathrm{m\,s^{-1}}$, $N_d = 40$ droplets in each influence sphere of the ice crystals at initial ambient saturation ratio $S_\infty = 0.847$ and droplet distance $\mathcal{D}_0 = 5R_i$.

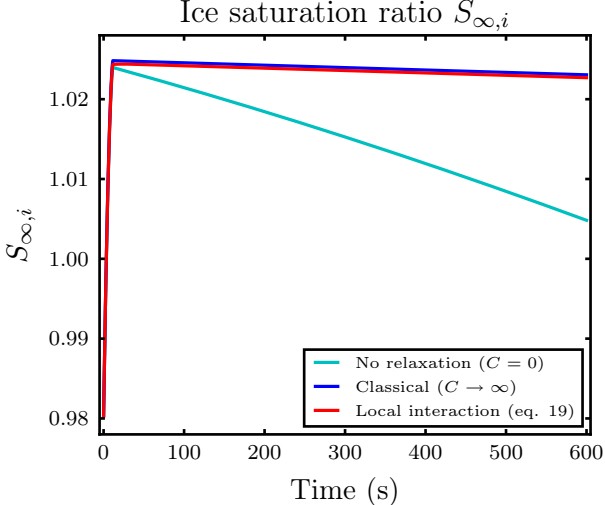

**Figure 8.** Temporal evolution of saturation ratio $S_{\infty,i}$ with respect to ice for the same simulation as in figure 7.





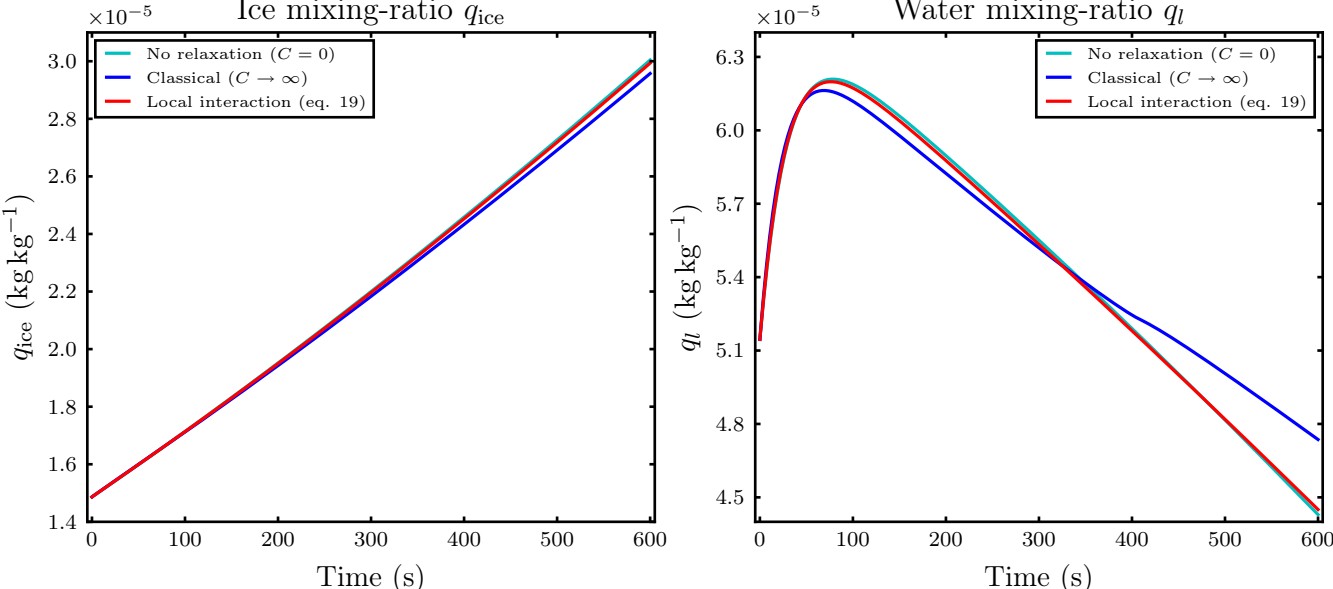

**Figure 9.** Parcel model simulation with $w = 0 \, \mathrm{ms}^{-1}$, $N_d = 500$ droplets in each influence sphere of the ice crystals at initial ambient saturation ratio $S_\infty = 1.01$ and droplet distance $\mathcal{D}_0 = 30 R_i$.

the evaporating ice crystals alleviate the local subsaturation, allowing the droplets inside the influence spheres to exist slightly longer, see case (c) in the schematic figure 1. The kink in the red curve for ice-mixing ratio in the left panel in figure 7 marks the time instant where all droplets are evaporated. From this time on, the local source for water vapor vanishes and the ice crystals grow slower. Note that the evaporated droplets provided enough water vapor to the whole air parcel to cause an ice

supersaturated environment, see figure 8, explaining why the ice crystals continue to grow although the air parcel was initially subsaturated with respect to ice.

Changing the environmental conditions to water supersaturation with $S_\infty = 1.01$, local effects on the mixing-ratios are almost not visible. Only for small distances as $\mathcal{D}_0 = 5 R_i$ of the droplets in the influence spheres from the ice crystals or very high droplet numbers $N_d$, an effect on the mixing-ratios is observed; for the case of a droplet–ice distance of $\mathcal{D}_0 = 30 R_i$ and

10 droplet number $N_d = 500$, see figure 9. The more interesting variable in this humidity regime is the temperature $T_\infty$ of the air parcel, shown in figure 10 for the aforementioned case. Compared to the classical treatment, including the local effects yields a slightly warmer air parcel. The heating is caused by the release of latent heat of the growing hydrometeors. It persists after $100 \, \mathrm{s}$ where the droplet mass starts to decrease and the evaporating droplets tend to cool the air parcel (right image in figure 9). Therefore, the observed heating is due to the growth of the ice crystals and should increase for increasing ice growth rate.

This motivates to consider the case of a small droplet–ice distance $\mathcal{D}_0 = 5 R_i$ and high droplet number $N_d = 500$ at ambient saturation ratio $S_\infty = 1.01$. The effect on the air parcel temperature $T_\infty$ for this case is shown in the left panel of figure 11. The right panel in figure 11 reveals a decrease in liquid water mixing-ratio already after a short time. This again confirms that the




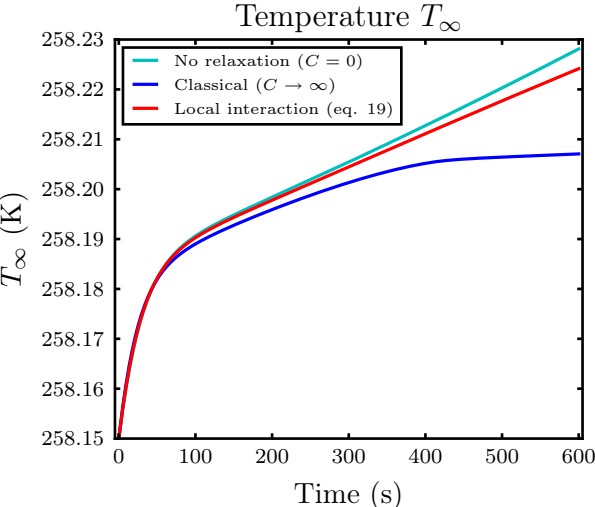

**Figure 10.** Temporal evolution of air parcel temperature $T_\infty$ for the same simulation as in figure 9.

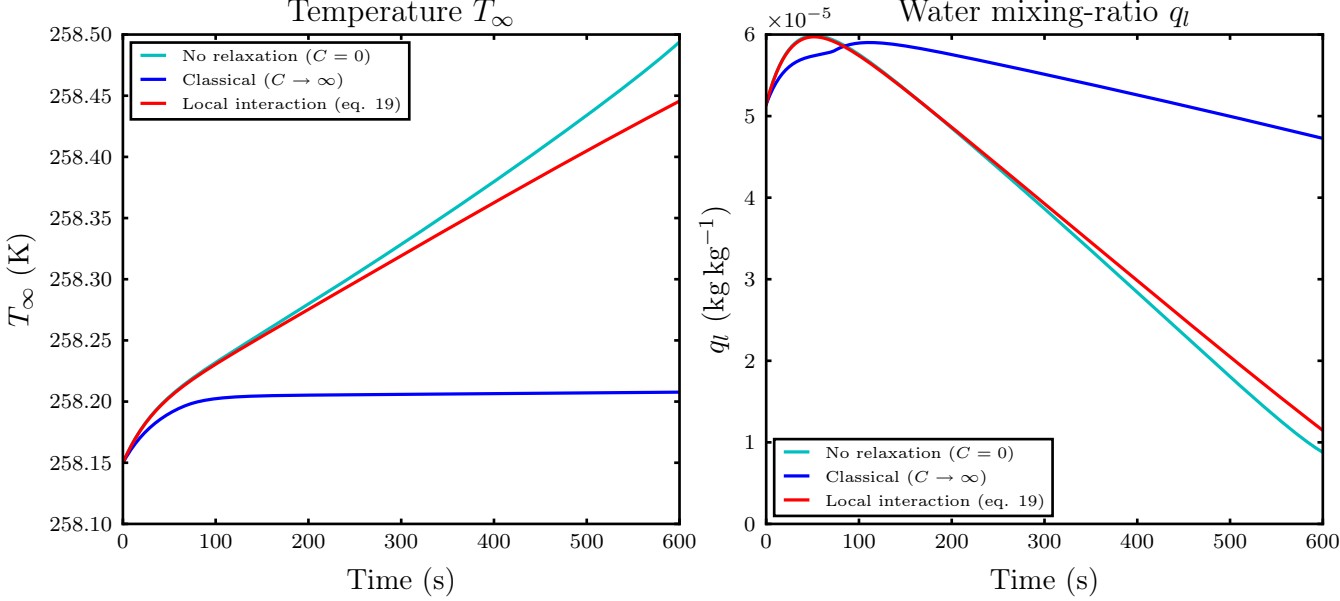

**Figure 11.** Parcel model simulation with $w = 0\,\mathrm{m\,s^{-1}}$, $N_d = 500$ droplets in each influence sphere of the ice crystals at initial ambient saturation ratio $S_\infty = 1.01$ and droplet distance $\mathcal{D}_0 = 5R_i$.

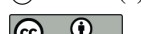


observed heating of the air parcel is caused by an increased growth rate of the ice crystals. The increase of the ice crystal growth rate is influenced by the number $N_d$ of droplets within the influence sphere and the ambient saturation ratio. If the air parcel is initially supersaturated with respect to water, the ice crystal induces a local subsaturation with respect to water and the droplets inside the influence sphere start to evaporate, see case (a) in the schematic in figure 1. Evaporation of nearby droplets enhances

the local water vapor density and consequently also the ice growth rate. If the air parcel is initially subsaturated with respect to water, the droplets inside the influence sphere of an ice crystal see two sinks of water vapor, namely the ice crystal and the environment, see case (b) in the schematic figure 1. The released water vapor of an evaporating droplet is therefore partially delivered to the ice crystal as well as the environment and the growth rate of the ice crystal is less amplified. Consequently, we expect a larger effect of the local interactions on the air parcel temperature with initially water supersaturated conditions.

### 3.2.2   Vertical Velocity $w = -1\,\mathrm{ms}^{-1}$

In a descending air parcel, saturation ratio decreases monotonically due to the adiabatic heating, causing an initially supersaturated air parcel to get subsaturated after a short time. According to the discussion in the previous subsection, we expect only negligible influence of the local effects on the temperature $T_\infty$ of the air parcel. This was confirmed by our conducted simulations.

Contrary to the case of vanishing vertical velocity discussed in the previous subsection, local effects are clearly visible in the mixing-ratios for an initially supersaturated air parcel. In the following, we show two examples, the first with a small droplet–ice distance and droplet number, the second with an increased droplet–ice distance and droplet number.

As the first example we choose a droplet–ice distance $\mathcal{D}_0 = 5R_i$, droplet number $N_d = 200$ and initial ambient saturation ratio $S_\infty = 1.01$. The temporal evolution of the mixing-ratios is shown in figure 12. From the ice mixing-ratio curve in the

left panel in figure 12 it is evident, that ice crystals evaporate much slower in comparison with the classical case without local interactions (blue curve). Also the droplets inside the influence spheres of ice crystals can exists longer compared to the classical case, see the right panel in figure 12. The droplets inside the influence sphere evaporate slower because their released water vapor raises the local coupling value $\rho_{v,*}$ and consequently slows down further evaporation. This explains the first kink of the red curve in the right panel in figure 12, where all outer droplets are evaporated. The second kink marks the complete

evaporation of all droplets. In this example, the droplets in the influence sphere exist up to $100\,\mathrm{s}$ longer than the droplets outside. As is indicated by the cyan curve, the time span may even be longer if the local relaxation rate $C$ is smaller. From the left panel in figure 12 we additionally observe that the red curve for ice mixing-ratio does not deviate significantly from the extreme case without local relaxation (cyan curve), until all droplets in the air parcel are evaporated. Therefore, a smaller local relaxation rate increases the time until the red curve deviates from the cyan curve.

In the second example with a moderate droplet–ice distance $\mathcal{D}_0 = 30R_i$, ambient saturation ratio $S_\infty = 1.01$ and $N_d = 500$ droplets, we observe similar effects as before, see figure 13. Using the larger droplet–ice distance it is important to have more droplet inside the influence spheres to compensate for the larger distance and to observe a similar effect of the local interactions. In this case, the delay in the complete evaporation of the droplets inside the influence spheres is about $50\,\mathrm{s}$, see the right panel in figure 13.




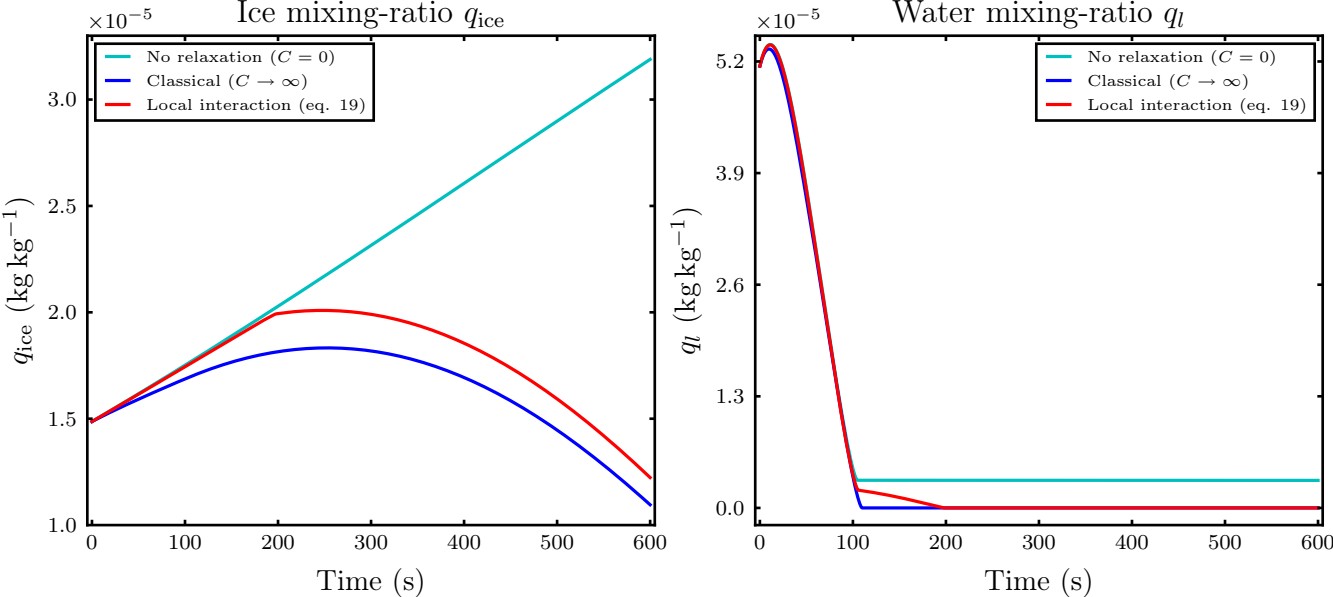

**Figure 12.** Parcel model simulation with $w = -1\,\mathrm{ms}^{-1}$, $N_d = 200$ droplets in each influence sphere of the ice crystals at initial ambient saturation ratio $S_\infty = 1.01$ and droplet distance $\mathcal{D}_0 = 5R_i$.

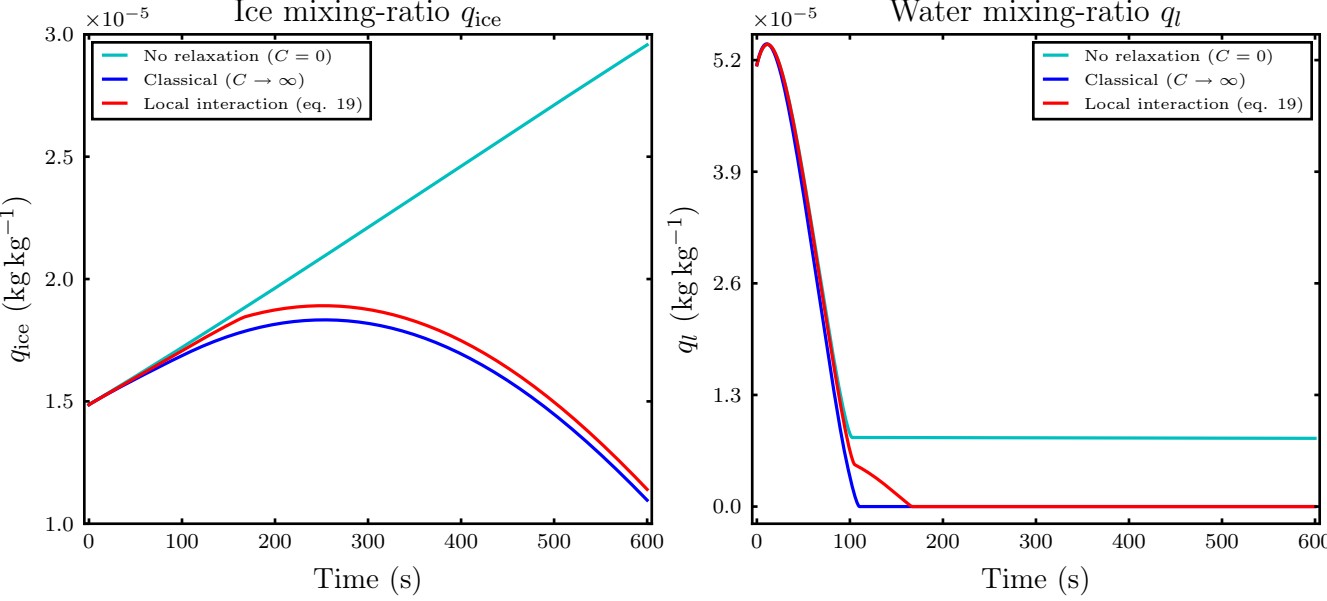

**Figure 13.** Parcel model simulation with $w = -1\,\mathrm{ms}^{-1}$, $N_d = 500$ droplets in each influence sphere of the ice crystals at initial ambient saturation ratio $S_\infty = 1.01$ and droplet distance $\mathcal{D}_0 = 30R_i$.




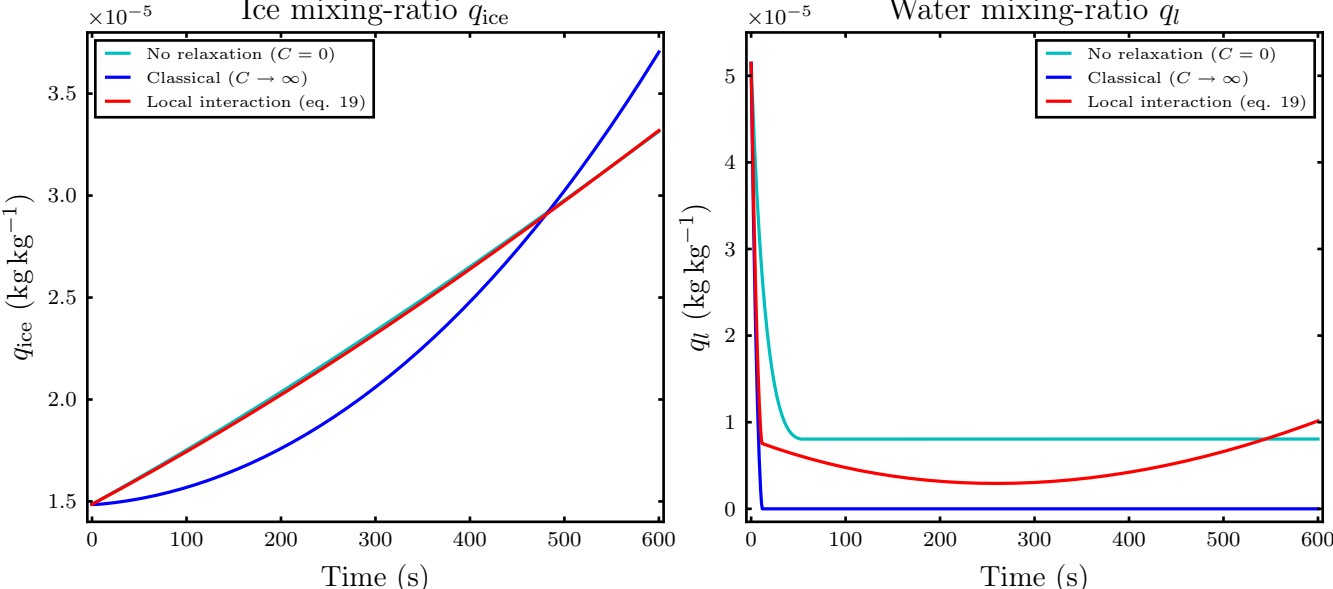

**Figure 14.** Parcel model simulation with $w = 1\,\mathrm{m\,s^{-1}}$, $N_d = 500$ droplets in each influence sphere of the ice crystals at initial ambient saturation ratio $S_\infty = 0.847$ and droplet distance $\mathcal{D}_0 = 5R_i$.

The longest delay of about $180\,\mathrm{s}$ in the evaporation of the droplets was found in the simulations using the small droplet–ice distance $\mathcal{D}_0 = 5R_i$ and initial saturation ratios $S_\infty \in \{0.932, 1.01\}$ (not shown).

### 3.2.3 Vertical Velocity $w = 1\,\mathrm{m\,s^{-1}}$

For updrafts, a significant effect of local interactions on the mixing-ratios was not observed in our conducted simulations. Even

in a massively water subsaturated regime $S_\infty = 0.847$ with small droplet–ice distance $\mathcal{D}_0 = 5R_i$ and high droplet number $N_d = 500$, where we expect the largest effect of the local interactions, an effect of the local interactions on the ice mixing-ratio was only minor, see the left panel in figure 14. Because of the ascend, the air parcel cools and the saturation ratio increases. Remarkably in this simulation, the droplets inside the influence spheres of the ice crystals managed to survive the time until the air parcel got saturated, whereas all droplets outside the influence spheres and in the classical treatment without the local

interactions evaporated earlier, see the right panel in figure 14. The air parcel got saturated with respect to water at about $200\,\mathrm{s}$, see figure 15. From the same figure it is evident, that the saturation ratio increased towards unrealistically high values of about $10\,\%$ after $350\,\mathrm{s}$, because we neglected activation of new droplets in our simulations. Such high supersaturations are in reality efficiently removed by activation and further diffusional growth of new droplets (Lamb and Verlinde, 2011, chapter 10).

However, considering only the simulations where the saturation ratio stayed within a reasonable realistic range, we show

as an example the case with initial water supersaturation and saturation ratio $S_\infty = 1.01$, droplet distance $\mathcal{D}_0 = 5R_i$ and $N_d = 500$ droplets per influence sphere. The temporal evolution of the mixing-ratios is shown in figure 16. Compared to the





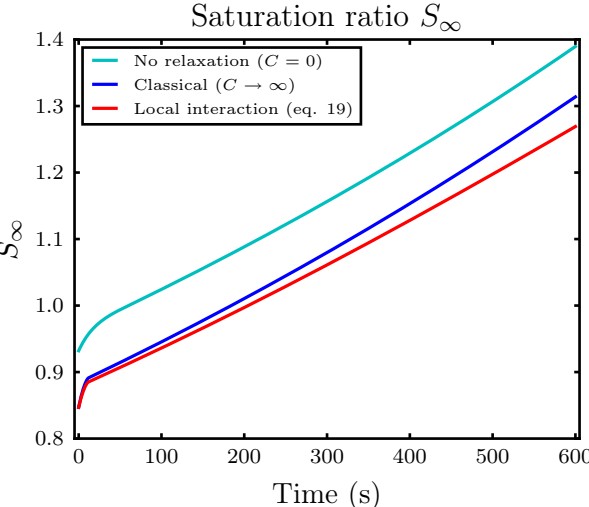

**Figure 15.** Temporal evolution of saturation ratio $S_\infty$ with respect to water for the same simulation as in figure 14.

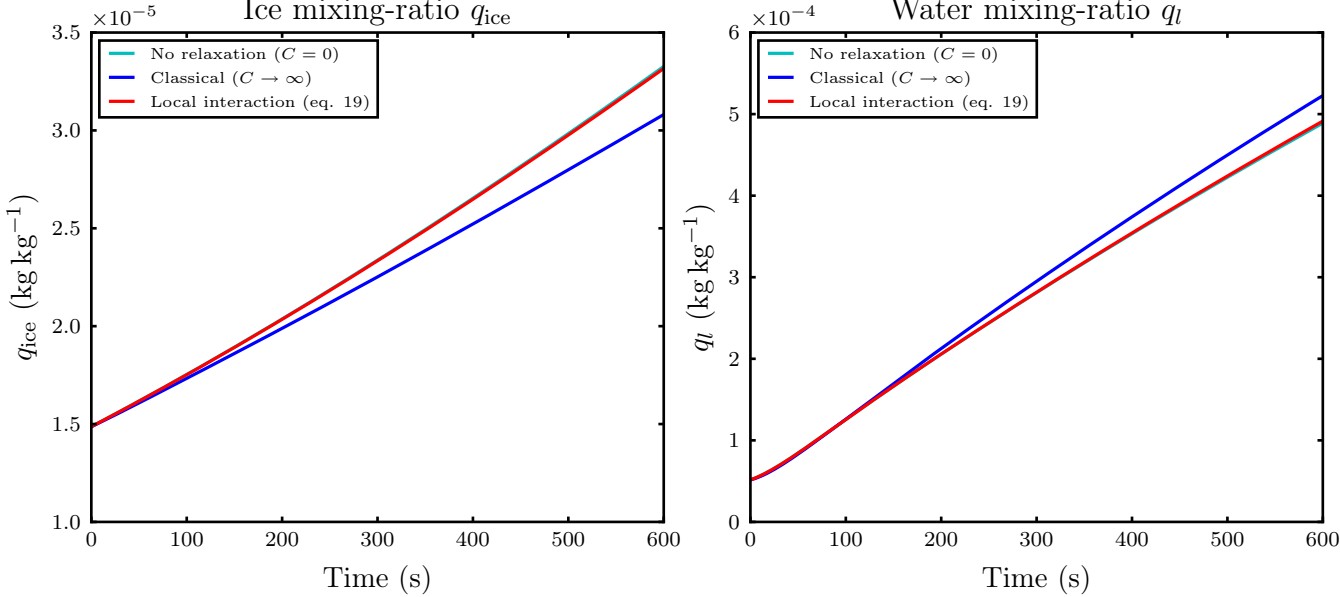

**Figure 16.** Parcel model simulation with $w = 1\,\mathrm{m\,s^{-1}}$, $N_d = 500$ droplets in each influence sphere of the ice crystals at initial ambient saturation ratio $S_\infty = 1.01$ and droplet distance $\mathcal{D}_0 = 5R_i$.





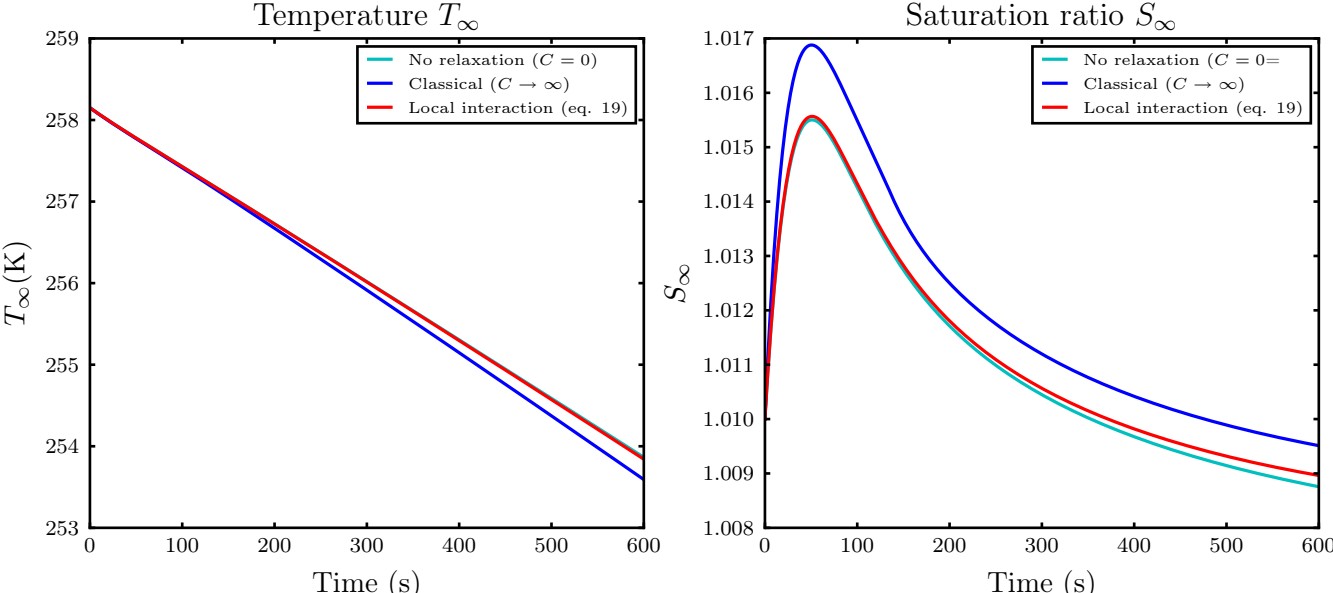

**Figure 17.** Temporal evolution of air parcel temperature $T_\infty$ and saturation ratio $S_\infty$ with respect to water for the same simulation as in figure 16.

classical case, the ice growth rate is slightly increased. This increase is sufficient to raise the temperature $T_\infty$ of the air parcel by approximately $0.25\,\mathrm{K}$ over the classical case, see the left panel in figure 17. From all conducted simulations with vertical velocity $w = 1\,\mathrm{ms}^{-1}$, this was the most pronounced effect on air parcel temperature.

## 4 Conclusions

In this study, we considered the modelling of local interactions between hydrometeors, specifically the case of an ice crystal and surrounding cloud droplets. We were interested in capturing the impact of locality on the diffusional growth of the hydrometeors. In contrast to the study by Baumgartner and Spichtinger (2017b), we suggested a formulation of the local interaction which may be suited to incorporate into a bulk microphysics model. Since this formulation allows a more physically consistent representation of the interaction between ice crystals and cloud droplets, the model may improve the representation of the

Wegener-Bergeron-Findeisen process. Apart from the derivation of the model, we incorporated the suggested model into an air parcel framework in order to assess the impact of local interactions on a mixed-phase cloud.

    The conducted simulations show that effects of local interactions increase for decreasing droplet–ice distances $\mathcal{D}_0$ and increasing number of droplets $N_d$ per influence sphere of the ice crystals. This dependence on the system parameters was to be expected by construction of the model in section 2 and is independent of the choice of the other parameters. All simulations

were carried out with initial radius $R_i = 100\,\mu\mathrm{m}$ for the ice crystal and $r_d = 10\,\mu\mathrm{m}$ for the cloud droplets. Other choices for




the initial radii will modify the rate of change in mass of the hydrometeors and therefore the overall intensity and duration of local effects but not the qualitative influence of the parameters $\mathcal{D}_0$ and $N_d$.

If the air parcel has no vertical velocity, local interactions enhance the growth of the ice crystals, especially in water subsaturated environments. This finding is in accordance with the qualitative results obtained with the reference model in Baumgartner

and Spichtinger (2017b) for ice- and water subsaturated environments. The effects of local interactions are clearly visible in contrast to the case of water supersaturated environments.

A descending air parcel, being initially supersaturated with respect to water, gets subsaturated with respect to water due to the adiabatic heating. The local interactions cause a delay in the evaporation of the hydrometeors. In order to observe a pronounced effect of the local interactions it is important, to have an air parcel being initially close to water saturation. In

this case, the surrounding droplets increase the local water vapor density and maintain a local higher saturation ratio. If the air parcel is initially subsaturated with respect to water the droplets inside the influence sphere evaporate more rapidly. Since also the environment is water subsaturated, a large portion of the released water vapor from the evaporating droplets diffuses into the environment due to the relaxation term in equation (19e). Therefore, the ice crystals can benefit only shortly from the released water vapor and the effects of the local interaction is smaller.

For an ascending air parcel, mostly no significant effects of local interactions on the mixing-ratios were observed in accordance with simulations of the reference model carried out with water subsaturated environmental conditions in Baumgartner and Spichtinger (2017b). The remarkable exception is illustrated in figure 14, where we observed a longer lifetime of the cloud droplets due to the local interaction with the ice crystal. In this particular case, the cloud droplets were able to survive a strong subsaturation with respect to water.

The effect of the local interactions is primarily controlled by the droplet–ice distance and the number of droplets in the influence spheres of the ice crystals. An enhancement of the local effects on the mixing-ratios is possible through a descending air parcel being initially close to saturation or supersaturated with respect to water. This conclusion is consistent with the theoretical study of Korolev (2008). In this study, various growth regimes of hydrometeors in a mixed-phase cloud were identified and connected to vertical velocities. In order to identify the different regimes, the author also employed a parcel model

with monodisperse size distributions of the hydrometeors, but excluded local interactions. It was shown, that the Wegener-Bergeron-Findeisen process is only active in downdrafts and has its maximal efficiency for vertical velocities around $w = 0\,\mathrm{ms}^{-1}$. Although these findings were obtained with an idealized air parcel model, they seem to be valid in general, because similar observations were made in large-eddy simulations of real clouds (Fan et al., 2011).

In our study, we additionally obtained an effect of the local interactions on the temperature $T_\infty$ of the air parcel. This

aspect is most pronounced for vanishing vertical velocities. If the air parcel is water supersaturated, the ice crystals induce a water subsaturation in their immediate vicinity and benefit from evaporating droplets located in this vicinity, see case (a) in the schematic figure 1. Consequently, the ice crystals enhance their growth rate and release more latent heat to the air parcel, causing the heating of the parcel. If the air parcel is initially water subsaturated, the enhancement of the ice growth rate is much less and the latent heating of the air parcel is diminished.




Since the heating of the air parcel through local interactions is due to an enhanced growth rate of the ice crystals, the strength of the temperature effect additionally depends on the number of ice crystals inside the air parcel. As detailed before, the ice crystal growth rate also depends on the number $N_d$ of droplets inside the influence spheres and the droplet–ice distance $\mathcal{D}_0$. If the air parcel has a non-vanishing vertical velocity, local interactions may influence the air parcel temperature only in the case

of an ascending parcel, for high droplet numbers inside the influence spheres and small droplet–ice distances.

One may speculate about the influence of a temperature change of the air parcel on its buoyancy which depends directly on the temperature of the parcel (Rogers and Yau, 1989, chapter 3). An additional heating of an air parcel with zero vertical velocity may trigger a vertical motion in an unstable stratification.

Although air parcel models are widely used, they might over- or underestimate the strength of observed effects. Therefore

one should include the suggested local-interaction model into a large-eddy model framework and again analyze the influence of the local interactions seen in this study with the more realistic model, which is left for future work.

In this study we only considered local interactions regarding the diffusional growth of the ice crystal and surrounding cloud droplets. Another aspect of hydrometeors with only small distances is an enhanced collision probability. It is known that small scale turbulence enhances collision probability. In our context, an enhanced collision probability means an enhanced

probability for riming of the ice crystals. In addition, the observed delays in the evaporation of the cloud droplets may also contribute to an increase in riming efficiencies.

*Acknowledgements.* We thank A. Seifert for fruitful discussions. This study was prepared with support by the German "Bundesministerium für Bildung und Forschung (BMBF)" within the HD(CP)[2] initiative, project M6 (001LK1207A).





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
