# Peer review of "Towards a Bulk Approach of Local Interactions of Hydrometeors"

_Atmospheric Chemistry and Physics, 2017_

## Referee Comment (RC1) · Anonymous Referee #1 · 24 Jul 2017

**Review of Towards a Bulk Approach of Local Interactions of Hydrometeors**

**General comments**

Baumgartner and Spichtinger derive a modified set of differential equations to describe diffusional growth of hydrometeors, one that incorporates local rather than far-field vapor density and temperature. They incorporate the new set of growth equations into a parcel model and illustrate the effects on evolution of water and ice mixing ratios and ambient temperature.

I think there are a number of interesting ideas throughout this work and that describing hydrometeor growth with local fields could ultimately be influential on mixed-phase microphysics in more complicated frameworks. But the model derivation and analysis need to be better explained before publication. For example, the coupling distance definition is not completely clear. On Page 4, Lines 10-11, you state that coupling values are those "at distance  $r_E$  from the droplet"; however, Figure 2, shows the coupling vapor density and temperature existing on a surface through the droplet center of mass. And in Equation 5, you define the coupling temperature existing at  $R_d$ . Would it help to define a coupling distance  $R_*$ , distinct from  $r_E$  or  $R_d$ , in Figures 2 and 3?

Then, I do not understand the notions of "smearing droplets" on the spherical shells or "bloating" these shells. On page 4, line 8, you state that "assuming spherical symmetric fields means that the droplet is smeared over the sphere with radius  $R_d$ ." Does this actually mean that the same vapor density is assumed for the whole shell as would exist at the surface of a single droplet? On Page 4, Line 12, you state that the sphere with coupling radius  $R_d$  is "bloated" to a shell with size  $2r_{E}$ , but presumably  $R_d$  is much larger than  $r_E$  (although  $r_E$  cannot be explicitly calculated). It could help to more thoroughly explain the significance of the gray region in Figure 3. In both instances, the wording needs to be more rigorous.

I am also still uncertain how saturation is evolved in time. It seems like this is done algebraically with the inline equations on Page 13, Line 1, rather than differentially in the parcel model. Is this correct? And the values listed in Equations 52 and 53 are initial values? Also all the parcel model simulations are performed for a single ice crystal, right? *N*ice in Equations 46 to 49c equals 1?

Clarification issues aside, I believe that the assumption of spherical ice crystals is an influential one and needs more discussion or rigorous justification than simply that "the vapor fields are very similar to those for a spherical ice crystal". You cite Lamb and Verlinde's Figure 8.24 to support this similarity, but they also describe the far field as being many crystal dimensions away, which is not the case for about half of the results you show in Section 3.2 when  $D_0 = 5R_i$ . These are also the simulations for which the effect of local interactions is most important, as you state on Page 25, Line 12. And studies like Sulia and Harrington 2011 *JGR* and Jensen and Harrington 2015 *JAS* have illustrated the large influence of aspect ratio on depositional growth of ice. The use of a saturation- and temperature-dependent deposition density as in Chen and Lamb 1994 *JAS* would also be more accurate than a bulk ice density.

Finally, I think dynamical effects could be better considered. Does the w = 0 case of Section 3.2.1 correspond to anything physically realistic? Would the large ice crystal not immediately sediment, obviating the establishment of any "spheres of influence"? Could hydrometeor sedimentation establish boundary layers that modify the local vapor densities? If the cloud parcel is turbulent, with droplets in

constant motion relative to the larger ice crystal, is the concept of a "sphere of influence" really meaningful? These kinds of questions should be considered in the conclusions, and particularly before any LES implementation.

**Specific comments**

Page 1, Abstract: The abstract would be made more compelling by explicitly stating what the "specific scenarios" are for which cloud droplets last longer.

Page 3, Line 29: Here you could go ahead and more rigorously define "non-negligible" crystal influence being when the relative deviation of the vapor density from the environmental value is greater than 0.1% as stated in Section 2.2.2.

Page 6, Equation 6 and Line 15: I think it makes more sense to immediately give the physical explanations of these three terms  $I_1$ ,  $I_2$ , and  $I_3$  after Equation 6 and then proceed to expand them mathematically. Otherwise, Equation 6 appears rather arbitrary.

Page 6, Lines 21-26: To me, it makes more sense to first write out the rate of water vapor exchange through the spherical surface (or just say water vapor flux) prior to Equation 3a and then only mention it again in the description of  $I_2$ .

Also, because you have already introduced the notion of "radii of influence", it seems clearer to me to define the water vapor flux with a double integral over this radius of influence and all angles, rather than introducing the notion of a "ball of influence" too and defining flux with a surface integral.

Page 6, Equation 9: I understand that you cannot simply assume that the gray shell in Figure 3 is filled completely with droplets; however, I do not understand the justification for instead assuming that it is filled with these droplets' "spheres of influences", as in this *Z* factor. Is there a physical reason that these "spheres of influences" could not overlap in-cloud?

Page 8, Lines 5-6: Given that *D* is generally a diffusion coefficient, it would be less confusing to use another variables for the positive constants in Equations 17a, 17b, and 18.

Page 9, Lines 20-22: It is a small difference, but I calculate  $D_0$  = 8.66 x 10-4 m. Have I missed something?

$$\begin{aligned} R_d &= \frac{\sqrt{3}}{2} \mathscr{L}_k = R_i + \mathscr{D}_0 \\ \frac{\sqrt{3}}{2} \frac{1}{\sqrt[3]{\mathcal{N}} - 1} &= 100 \times 10^{-6} \text{ m} + \mathscr{D}_0 \\ \frac{0.866}{\sqrt[3]{\mathcal{N}} - 1} &= 100 \times 10^{-6} \text{ m} + \mathscr{D}_0 \\ \mathscr{D}_0 &= \frac{0.866}{\sqrt[3]{1000} \text{ cm}^{-3} - 1} - 100 \times 10^{-6} \text{ m} \\ &= 0.866 \left(\frac{1}{9}\right) \times 10^{-2} \text{ m} - 100 \times 10^{-6} \text{ m} \end{aligned}$$

Page 9, Equation 23 and Page 10, Lines 10 and 11: If you are already using the far field pressure and temperature to calculate the surface water vapor density in Equations 23 and 28, what do you use to

calculate the far field water vapor density? Does the temporal derivative of the far field vapor density in Equation 20 come from the Clausius Clapeyron equation also?

Section 2.2.4 and Figures 5 and 6: I do not see where n = 1.8 appears in these figures. You need to clarify why and how the droplet and ice mass evolutions allow you to derive this parameter. As a side note, you spent Section 2.2.3 justifying the use of  $N_d = 40$ . Why don't you use this value in this section rather than the seemingly arbitrary values of 14 and 38?

Page 14, Lines 1-3 and 11: Will it actually be possible to make the necessary measurements to constrain parameters or assess the validity of the Z factor? You might mention the apparatus or technique that would be appropriate to measure vapor densities at the requisite spatial scales.

Page 15, Lines 20-27: This paragraph needs clarification. Introduce all the number variables first  $N_i$ ,  $n_d^j$ ,  $n_d^o$ . Then introduce all the mass variables  $M_{ice}$ ,  $m_d^j$ ,  $m_d^o$  and finally the temperatures  $T_i T_d^j$ ,  $T_d^o$ .

At the moment, I am unclear why the liquid mass variables are introduced as  $M_i^j$  and  $M_i^o$  here, as they have been denoted  $m_d$  earlier and appear this way in Equation 46 also. Extending this point, although it is a detail, it will help with clarity if you use consistent subscripts for the liquid phase, either *I* or *d*, and the ice phase, either *ice* or *i*, throughout. It would also be helpful to consistently use the upper case letters for the ice crystal variables and lower case ones for the droplet variables, as you mention early on in the derivation.

Page 16, Line 12 to Page 17, Line 2: It would be easier to read this list of simulations in a table format.

Page 16, Lines 17-20: I understand that these three values of saturation ratio are chosen as the three cases shown in Figure 1. Is there any other reason for the exact values chosen? i.e., why  $S_{\infty}$  = 0.847 rather than simply 0.8 or 0.9? Are the results quite particular to the  $S_{\infty}$  value?

Page 17, Line 6: Why is  $N_d$  limited even to [30, 800]?

Page 17, Lines 7-9: The purpose of the list of citations is not clear. Please reword, perhaps like "*Previous studies indicate a large scattering of microphysical parameters, especially LWC and IWC [Fleishauer et al., Hobbs et al., Noh et al., ...]. We use typical values ..."* Then what do you mean by "typical values" for 0.045 and 0.013 g m-3? These are mean or mode values from the cited observations?

Page 18, Figure 7: Really, the x-axis here does not need to run for 10 minutes because all the activity that you discuss occurs within the first 100 seconds. I think that you have included the full axis to be consistent with Figures 9, 11, etc. If this is the motivation, I think the clearest way to show results for section 3.2.1 would be a 2-by-3 figure where you show  $q_i$  and  $q_j$  for the  $N_d = 40 + D_0 = 5R$ ,  $N_d = 500 + D_0 = 30R$ , and  $N_d = 500 + D_0 = 5R$  simulations, i.e., combine Figures 7, 9, and 11 to ease comparison. Show the temperature fields separately.

Page 18, Figure 7 and Section 3.2.1 discussion: It would be informative to include the timing of the "kinks". At what time does your local interaction model stop following the no relaxation one in the left hand panel

of Figure 7? You could also include time units on the Figure 7 inset. What is the additional time for which "droplets inside the influence sphere exist"?

Page 19, Figure 9 right hand panel and Page 20 Figure 10: What causes the kink at about 400 s in the classical model?

Page 22, Figures 12 and 13: I think these figures could also be combined into a 2-by-2 panel to ease comparison at  $w = -1 \text{ m s}^{-1}$ .

Conclusions section: I think it would help readers understand the various trends you simulate if you organize them in a small table. Something like

|                | Stationary             | Ascending | Descending |
|----------------|------------------------|-----------|------------|
| S∞             | Effects most important |           |            |
|                | for subsaturation      |           |            |
| T∞             |                        |           |            |
| N d |                        |           |            |
| $D_0$          | etc.                   |           |            |

**Technical comments / suggestions**

Page 1, Line 24: "much more severe impact"

Page 4, Line 14: "However, the idea is as follows": This transition does not seem natural. You are not contradicting anything that just proceeded.

Page 7, Lines 24-25: Reword for clarity. "Since V is the volume into which vapor diffuses in unit time."

Page 10, Line 26: "all droplets with distances smaller than Rd/3 have larger influence"

Page 11, Lines 8-9: This is a run-on sentence. "the artificial spherical shell. If n > 1, only a fraction  $n^{-3}$  is incorporated, and the remaining water vapor is released to the atmosphere". Also it would be better to say "affects the coupling values" than to say "is incorporated".

Page 13, Line 23: "where the authors also consider"

Page 15, Line 7: Better to say "normal to the surface of the hydrometeor  $\delta w_k$ . The surface integral..."

Page 16, Line 9: Remove the "503.61377pt" text.

Page 17, Line 14: "In the subsequent sections"

Page 18, Figure 8: I would suggest to move Figure 8 to supplemental information for conciseness. It is mentioned once briefly.

---

## Referee Comment (RC2) · Anonymous Referee #2 · 26 Jul 2017

**1   General Comments**

This article deals with models for the evolution of water droplets and ice crystals in clouds where bother kinds of hydrometeor are present. The individual modelling steps are mostly well explained, although I have a few comments below. However, the overall structure is not as clearly presented - the discussion at the start of S2.1 leads straight into moderately complex equations. This referee would appreciate some abstract form of the model with terms that are then explained/defined/expanded in following subsections.

My research focuses on particles that are not hydrometeors, so it is difficult for me to assess exactly how significant the progress made in this article is. My impression is that

it is a useful next step in extending the models available for large scale computations, but the results in the paper are not particularly remarkable in themselves.

**2 Specific Comments**

p1 l26 I would appreciate a comment on the relation to Ostwald ripening of crystals/emulsions.

p4 l12 Turning the surface of a sphere with radius $R_d$ into a shell of thickness $2r_E$ seems to be more or less equivalent to the assumption that $r_E$ is small compared to $R_d$. Please comment on why this key assumption is reasonable.

p6 l27 "using the exchange rate $J_d/V$" directly" cannot amount to anything since $V$ is still to be defined. I think the explanation at this point could be improved. $J_d$ is a flux, if we assume that the shell around the ice crystal remains well mixed, then naturally the change in concentration is the flux into the shell divided by the volume of the shell and if there are multiple fluxes they should be summed (which here reduces to multiplying by $N_d$). My suggestion is to eliminate both $V$ and $Z$.

p8 l2 Please explain why $n$ cannot be estimated from the diffusion constant of water vapour. Effectively $n$ seems to be a fitting parameter for the model, which is fine, but it would be helpful to know why n and not some other quantity should be a free parameter.

p27 I think the conclusion is much too long, because a large number of observations are repeated. I suggest stressing the most important one or two observations, discussing the prospects for future work and not taking more than half of a page!

**3 Minor remarks**

p1 l25 "it may evaporate nearby droplets and grow at their expense". Please rephrase, one cannot use evaporate like this. Perhaps "... may accelerate the evaporation of nearby droplets by growing at their expense".

p2 What is "ice saturation" please?

p2 Figure 1 should be made far more compact.

p4 l1 What are "objective estimates", perhaps measurement based ...

p4 l2 about -> around

p4 l12 I suggest avoiding bloat/bloated in physical science writing, perhaps "expand" or "dilate" would be suitable.

p7 eq15 seems to be a trivial rearrangement. I think it is safe to assume that readers can calculate the volumes of spheres and spherical shells.

p9 Figure 4 is superfluous, I suggest just writing a perfectly regular distribution of droplets at the vertices of a cubic lattice.

p14 l14 A slightly more detailed description of the parcel model would help this referee. I think we are talking about a parcel of gas moving in the overall flow field.

p14 l22 It is not clear to me exactly what $w$ is. Is it the vertical velocity of the atmospheric flow field at the centre of the air parcel?

p21 l12 "get subsaturated", better style would be "become subsaturated". This comment applies throughout the manuscript.

p21 l21 "exists" -> exist p21 l32 "droplet" -> droplets

2017.

---

## Author Comment (AC1) · 30 Aug 2017

Please see the general author's response in supplementary file.

---

## Author Comment (AC2) · 30 Aug 2017

Please see the general author's response in supplementary file.
* * *

---

## Author Comment (AC3) · 30 Aug 2017

**Response to referee comments**

Manuel Baumgartner[1,2] and Peter Spichtinger[1]

[1] *Institute for Atmospheric Physics, Johannes Gutenberg University, Mainz, Germany*
[2] *Zentrum für Datenverarbeitung, Johannes Gutenberg University, Mainz, Germany*

August 30, 2017

**Contents**

**1 General Response**

First of all we would like to thank both reviewers for their helpful comments and suggestions, which lead to an improvement of our manuscript.

Following the general suggestions of the referees, we split subsection 2.1 into two parts. The first part contains a general description of the modelling approach while the second part contains the derivation of the model equations. The text of the general description is largely rewritten for clarifying the presentation and to provide an overview of the modelling idea. Furthermore we eliminated the former notions of "smearing" and "bloating". Following a suggestion of referee #1, we changed the notation of the coupling distance from $R_d$ to $R_*$, to ensure a consistent notation with the coupling values $\rho_{v,*}$ and $T_*$. We also changed the notations of the positive constants $\mathcal{D}_i$, $\mathcal{D}_d$, $\mathcal{D}_0$ to $l_i$, $l_d$ and $l_0$ to avoid confusion with the diffusion coefficients.

Subsection 3.2 now contains a table to summarize our choices for the parameters for the air parcel simulations instead of enumerations of the parameters within the main text. We combined several individual figures into single figures with multiple rows. The old figures 7, 9 and the right panel in the old figure 11 are now combined into the new figure 6. We also combined the temperature plots of the old figures 10 and 11 into the new figure 7. Similarly, we combined the old figures 12, 13 into the new figure 8 and the old figures 14, 16 into the new figure 9. As suggested by referee #2, we deleted the old figure 4. Following a suggestion by referee #1, we moved the old figure 8 into the appendix as the new figure B1.

**2 Response to Referee #1**

**2.1 Response to General Comments**

- I think there are a number of interesting ideas throughout this work and that describing hydrometeor growth with local fields could ultimately be influential on mixed-phase microphysics in more complicated frameworks. But the model derivation and analysis need to be better explained before publication. For example, the coupling distance definition is not completely clear. On Page 4, Lines 10-11, you state that coupling values are those "at distance $r_E$ from the droplet"; however, Figure 2, shows the coupling vapor density and temperature existing on a surface through the droplet center of mass. And in Equation 5, you define the coupling temperature existing at $R_d$. Would it help to define a coupling distance $R_*$ , distinct from $r_E$ or $R_d$, in Figures 2 and 3?

As mentioned in the general comments, we replaced the notation $R_d$ for the coupling distance in favor of the notation $R_*$. The coupling distance $R_*$ is defined through the distance of the droplet from the ice crystal center and we assume the fields of water vapor and temperature to equal the coupling values $\rho_{v,*}$ and $T_*$ at this distance from the ice crystal. Due to the spatial position of the droplet, these values are the relevant values for the diffusional growth of the droplet.

Since the Maxwellian growth equations (3) for the droplet growth need "environmental values" to compare them with the values on the droplet's surface $\rho_{v,d}$ and $T_d$, we need the concept of the influence sphere for the droplets too. For the coupling of the diffusional growth of the ice crystal and the droplet, we assume that the "environmental values" for the droplet growth, i.e. the values at the boundary of the droplet's influence sphere, equal the coupling values. The old figure 2 is misleading in this respect, so we deleted the indication of the droplet influence sphere in the new version.

- Then, I do not understand the notions of "smearing droplets" on the spherical shells or "bloating" these shells. On page 4, line 8, you state that "assuming spherical symmetric fields means that the droplet is smeared over the sphere with radius $R_d$." Does this actually mean that the same vapor density is assumed for the whole shell as would exist at the surface of a single droplet? On Page 4, Line 12, you state that the sphere with coupling radius $R_d$ is "bloated" to a shell with size $2r_E$, but presumably $R_d$ is much larger than $r_E$ (although $r_E$ cannot be explicitly calculated). It could help to more thoroughly explain the significance of the gray region in Figure 3. In both instances, the wording needs to be more rigorous.

We rewrote the corresponding parts of the manuscript to clarify those issues.

In our model we deal with vapor densies, so we also have to define a vapor density along the sphere with coupling radius $R_*$ around the ice crystal. Since a two dimensional sphere has volume zero in $\mathbb{R}^3$, we have to expand this sphere into a three dimensional manifold with positive volume in order to define a density. This motivates the expansion of the sphere into a spherical shell. Since vapor density along the sphere is assumed to equal the coupling value $\rho_{v,*}$, we assume to have the same value uniformly inside the expanded, artificial spherical shell.

As you stated correctly, $R_*$ is larger than $r_E$. Both values are connected to the sizes of the hydrometeors and it typically, an ice crystal is larger than a droplet.

- I am also still uncertain how saturation is evolved in time. It seems like this is done algebraically with the inline equations on Page 13, Line 1, rather than differentially in the parcel model. Is this correct? And the values listed in Equations 52 and 53 are initial values? Also all the parcel model simulations are performed for a single ice crystal, right? $\mathcal{N}_{\text{ice}}$ in Equations 46 to 49c equals 1?

  In the parcel model simulations, the mixing-ratio $q_v$ for water vapor is evolved using the differential equation (48) and afterwards converted into the corresponding value for saturation ratio. All values listed in the old equations (52) and (53) are now listed in the new table 1 and are indeed initial values for the air parcel. For the ice crystals we assume a monodisperse size distribution with number density $\mathcal{N}_{\text{ice}}$, where the number density is calculated using the given value $\text{IWC} = 0.013\,\text{g}\,\text{m}^{-3}$ for the ice water content. This leads to $\mathcal{N}_{\text{ice}} \approx 3.861 \cdot 10^3\,\text{kg}^{-1}$.

- Clarification issues aside, I believe that the assumption of spherical ice crystals is an influential one and needs more discussion or rigorous justification than simply that "the vapor fields are very similar to those for a spherical ice crystal". You cite Lamb and Verlinde's Figure 8.24 to support this similarity, but they also describe the far field as being many crystal dimensions away, which is not the case for about half of the results you show in Section 3.2 when $D_0 = 5R_i$. These are also the simulations for which the effect of local interactions is most important, as you state on Page 25, Line 12. And studies like Sulia and Harrington 2011 JGR and Jensen and Harrington 2015 JAS have illustrated the large influence of aspect ratio on depositional growth of ice. The use of a saturation- and temperature-dependent deposition density as in Chen and Lamb 1994 JAS would also be more accurate than a bulk ice density.

  Our assumption of spherical ice crystals is indeed an influential one and we now comment more on this issue: In the main text we clarified the impact of this assumption and we added an appendix to illustrate the effect of the shape of the ice crystal on the diffusional growth of surrounding droplets.

- Finally, I think dynamical effects could be better considered. Does the $w = 0$ case of Section 3.2.1 correspond to anything physically realistic? Would the large ice crystal not immediately sediment, obviating the establishment of any "spheres of influence"? Could hydrometeor sedimentation establish

boundary layers that modify the local vapor densities? If the cloud parcel is turbulent, with droplets in constant motion relative to the larger ice crystal, is the concept of a "sphere of influence" really meaningful? These kinds of questions should be considered in the conclusions, and particularly before any LES implementation.

The case of an air parcel with vertical velocity of precisely zero is probably a rare case in reality, but we think of this case as the limiting case of very small vertical velocities, occuring much more frequent. However, this case illustrates best the local effects without masking them with another effects arising from dynamics.

We added in section 2.4 more comments about the other remarks, specifically what are the implications of a non-constant droplet number $N_d$ and how to define the sphere of influence for a moving ice crystal.

**2.2 Response to Specific Comments**

**Page 1, abstract** The abstract would be made more compelling by explicitly stating what the "specific scenarios" are for which cloud droplets last longer.

We now mention case of a water supersaturated air parcel within a downdraft in the abstract.

**Page 3, line 29** Here you could go ahead and more rigorously define "non-negligible" crystal influence being when the relative deviation of the vapor density from the environmental value is greater than $0.1\,\%$ as stated in Section 2.2.2.

We followed the suggestion and added our definition of "non-negligible" from section 2.2.2.

**Page 6, equation 6 and line 15** I think it makes more sense to immediately give the physical explanations of these three terms $I_1, I_2$, and $I_3$ after Equation 6 and then proceed to expand them mathematically. Otherwise, Equation 6 appears rather arbitrary.

We added a list where we give a short explanation of the terms.

**Page 6, lines 21-26** To me, it makes more sense to first write out the rate of water vapor exchange through the spherical surface (or just say water vapor flux) prior to Equation 3a and then only mention it again in the description of $I_2$. Also, because you have already introduced the notion of "radii of influence", it seems clearer to me to define the water vapor flux with a double integral over this radius of influence and all angles, rather

than introducing the notion of a "ball of influence" too and defining flux with a surface integral.

We moved the calculation of the surface flux $J_d$ right after equation (3). To us, a notation for a surface integral involving an integration over all angles seems far more complicated compared to a standard integral over the surface of a manifold with the appropriate surface measure $d\sigma$, so we stick with the old notation.

**Page 6, equation 9** I understand that you cannot simply assume that the gray shell in Figure 3 is filled completely with droplets; however, I do not understand the justification for instead assuming that it is filled with these droplets' "spheres of influences", as in this Z factor. Is there a physical reason that these "spheres of influences" could not overlap in-cloud?

We clarified this point in the main text and pointed out, that the precise form of the factor $Z$ is our choice.

**Page 8, lines 5-6** Given that $D$ is generally a diffusion coefficient, it would be less confusing to use another variables for the positive constants in Equations 17a, 17b, and 18.

We changed $\mathcal{D}_i$, $\mathcal{D}_d$ and $\mathcal{D}_0$ to $l_i$, $l_d$ and $l_0$, see the general response.

**Page 9, lines 20-22** It is a small difference, but I calculate $D_0 = 8.66 \cdot 10^{-4}\,\mathrm{m}$. Have I missed something?

The difference stems from the "1" in the denominator of equation (25), where we did not indicate the precise unit. Now we indicated the correct unit, which forces to use $\mathcal{N}$ with unit $\mathrm{m}^{-3}$ instead of $\mathrm{cm}^{-3}$. This caused the difference.

**Page 9, equation 23 and page 10, lines 10 and 11** If you are already using the far field pressure and temperature to calculate the surface water vapor density in Equations 23 and 28, what do you use to calculate the far field water vapor density? Does the temporal derivative of the far field vapor density in Equation 20 come from the Clausius Clapeyron equation also?

The far field values are given by the initial conditions. The pressure $p_{i,\infty}$ in the old equation (23) denotes the saturation pressure over a plane ice surface. To avoid further confusion, we replaced this by the notation $p_{i,\mathrm{sat}}$ and also replaced the saturation pressure $p_{l,\infty}$ over a plane surface of liquid water by the notation $p_{l,\mathrm{sat}}$ throughout the manuscript.

The computation of the derivative in equation (20) does not employ the Clausius-Clapeyron equation. It is calculated by differentiating equation (1a) with respect to $R$ and substituting $R = R_*$.

**Section 2.2.4 and figures 5 and 6** I do not see where $n = 1.8$ appears in these figures. You need to clarify why and how the droplet and ice mass evolutions allow you to derive this parameter. As a side note, you spent Section 2.2.3 justifying the use of $N_d = 40$. Why don't you use this value in this section rather than the seemingly arbitrary values of 14 and 38? We clarified that the given figures do not allow to deduce the value $n = 1.8$, they only show the accordance of the temporal evolutions of droplet mass and ice mass for the reference model and the for the new model using the parameter value $n = 1.8$.

Our use of the droplet numbers 14 and 38 is motivated by the Lebedev quadrature formulas for spheres (Lebedev, 1976), for which the respective integration points are nearly uniformly distributed along the sphere. We used those integration points as centers for the droplets to distribute them nearly uniformly around the ice crystal. In the main text we added a suitable comment.

**Page 14, lines 1-3 and 11** Will it actually be possible to make the necessary measurements to constrain parameters or assess the validity of the $Z$ factor? You might mention the apparatus or technique that would be appropriate to measure vapor densities at the requisite spatial scales.
In our opinion, at the moment it is not possible to make the necessary measurements since it remains a challenge to track a given air volume. We believe that eventually the technique of holographic imaging may help to make measurements to constrain some parameters. In section 2.4, we added a comment.

**Page 15, lines 20-27** This paragraph needs clarification. Introduce all the number variables first $N_i$, $n_d^i$, $n_d^o$. Then introduce all the mass variables $M_{\text{ice}}, m_d^i, m_d^o$ and finally the temperatures $T_i, T_d^i, T_d^o$. At the moment, I am unclear why the liquid mass variables are introduced as $M_l^i$ and $M_l^o$ here, as they have been denoted $m_d$ earlier and appear this way in Equation 46 also. Extending this point, although it is a detail, it will help with clarity if you use consistent subscripts for the liquid phase, either $l$ or $d$, and the ice phase, either ice or $i$, throughout. It would also be helpful to consistently use the upper case letters for the ice crystal variables and lower case ones for the droplet variables, as you mention early on in the derivation.
We rewrote large parts to clarify these issues. The motivation is to distinguish between a single droplet (index "$d$") and all the liquid droplets inside the air parcel (index "$l$"). Similarly, index "$i$" indicates a single ice

crystal while index "ice" indicates all ice crystals. We corrected an error in equations (46), (48) and (49a), where $M_{\mathrm{ice}}$ should read $M_i$, the mass of the individual ice crystal.

**Page 16, line 12 to page 17, line 2** It would be easier to read this list of simulations in a table format.
We followed this suggestion.

**Page 16, lines 17-20** I understand that these three values of saturation ratio are chosen as the three cases shown in Figure 1. Is there any other reason for the exact values chosen? i.e., why $S_\infty = 0.847$ rather than simply 0.8 or 0.9? Are the results quite particular to the $S_\infty$ value?
The chosen values are not particular in any respect. The value $S_\infty = 0.847$ comes from the choice $S_{\infty,i} = 0.98$, ensuring an ice subsaturated environment. The pair $(S_\infty, S_{\infty,i}) = (0.932, 1.076)$ ensures an intermediate regime between ice saturation and water saturation.

**Page 17, line 6** Why is $N_d$ limited even to $[30, 800]$?
This is motivated by the estimations from equations (30) and (31) in subsection 2.2.3.

**Page 17, lines 7-9** The purpose of the list of citations is not clear. Please reword, perhaps like "Previous studies indicate a large scattering of microphysical parameters, especially LWC and IWC [Fleishauer et al., Hobbs et al., Noh et al., ...]. We use typical values ..." Then what do you mean by "typical values" for 0.045 and $0.013\,\mathrm{g\,m^{-3}}$? These are mean or mode values from the cited observations?
We reformulated the sentence as suggested. The value for LWC is obtained by visual inspection of figure 7 in Korolev et al. (2003). Similarly, the value for IWC is a choice according to the observational campaign described in Fleishauer et al. (2002). Our choice of this value is within the range given in Fleishauer et al. (2002) and also consistent with figure 6 in Korolev et al. (2003).

**Page 18, figure 7** Really, the x-axis here does not need to run for 10 minutes because all the activity that you discuss occurs within the first 100 seconds. I think that you have included the full axis to be consistent with Figures 9, 11, etc. If this is the motivation, I think the clearest way to show results for section 3.2.1 would be a 2-by-3 figure where you show $q_i$ and $q_l$ for the $N_d = 40 + D_0 = 5R$, $N_d = 500 + D_0 = 30R$, and $N_d = 500 + D_0 = 5R$ simulations, i.e., combine Figures 7, 9, and 11 to ease comparison. Show

the temperature fields separately.
We followed the suggestion, see the general response.

**Page 18, figure 7 and section 3.2.1 discussion** It would be informative to include the timing of the "kinks". At what time does your local interaction model stop following the no relaxation one in the left hand panel of Figure 7? You could also include time units on the Figure 7 inset. What is the additional time for which "droplets inside the influence sphere exist"?
We included the timings of the described kinks in our figures. However, we did not add the time units in the inset in the upper row of figure 6 due to clarity.

**Page 19, figure 9 right hand panel and page 20 Figure 10** What causes the kink at about $400\,\mathrm{s}$ in the classical model?
We used our new model formulation also for the limiting cases with $C = 0$ and $C \to \infty$. Therefore, also for the "classical" computation we divided the droplets inside the air parcel into $q_l^i$ and $q_l^o$. At this kink, the value of $q_l^i$ reaches zero and the curve for $q_l = q_l^i + q_l^o$ follows solely the evolution of $q_l^o$.

**Page 22, figures 12 and 13** I think these figures could also be combined into a 2-by-2 panel to ease comparison at $w = -1\,\mathrm{m\,s^{-1}}$.
We followed the suggestion, see the general response.

**Conclusions section** I think it would help readers understand the various trends you simulate if you organize them in a small table. Something like
We followed the suggestion, see the general comments.

**2.3 Response to Technical Comments**

**Page 1, line 24** "much *more* severe impact"
We added the word "more".

**Page 4, line 14** "However, the idea is as follows": This transition does not seem natural. You are not contradicting anything that just proceeded.
We deleted "however".

**Page 7, lines 24-25** Reword for clarity. "Since $V$ is the volume into which vapor diffuses in unit time."
We followed the suggestion.

**Page 10, line 26** "all droplets with distances smaller than $R_d/3$ have larger influence"
We followed the suggestion.

**Page 11, lines 8-9** This is a run-on sentence. "the artificial spherical shell. If $n > 1$, only a fraction $n^{-3}$ is incorporated, and the remaining water vapor is released to the atmosphere". Also it would be better to say "affects the coupling values" than to say "is incorporated".
We followed the suggestions.

**Page 13, line 23** "where the authors also consider"
We added the word "where".

**Page 15, line 7** Better to say "normal to the surface of the hydrometeor $\delta w_k$. The surface integral..."
We followed the suggestion.

**Page 16, line 9** Remove the "503.61377pt" text.
This was produced by an unnecessary LaTeX-command. We removed it.

**Page 17, line 14** "In the subsequent sections"
We replaced "sequel" by "subsequent".

**Page 18, figure 8** I would suggest to move Figure 8 to supplemental information for conciseness. It is mentioned once briefly.
We moved this figure to the appendix, see the general response.

**3 Response to Referee #2**

**3.1 Response to General Comments**

- However, the overall structure is not as clearly presented - the discussion at the start of S2.1 leads straight into moderately complex equations. This referee would appreciate some abstract form of the model with terms that are then explained/defined/expanded in following subsections.
  We split section 2.1 into two parts, see the general response. As mentioned there, we rewrote large parts to make the presentation clearer. We added comments to the abstract form of the water vapor coupling equation (7) to explain the three terms $I_1$, $I_2$, $I_3$ shortly.

**3.2 Response to Specific Comments**

**Page 1, line 26** I would appreciate a comment on the relation to Ostwald ripening of crystals/emulsions.
We included a comment stating the difference of the Wegener-Bergeron-Findeisen process to Ostwald ripening.

**Page 4, line 12** Turning the surface of a sphere with radius $R_d$ into a shell of thickness $2r_E$ seems to be more or less equivalent to the assumption that $r_E$ is small compared to $R_d$. Please comment on why this key assumption is reasonable.
The values $R_*$ and $r_E$ are both connected to the size of the hydrometeors. Typically, an ice crystal is larger than a cloud droplet by a factor 10 or even more. Therefore, using the estimates given in section 2.2, we arrive at $r_E = r_d + l_d = 10r_d \approx 100\,\mu m$ for a typical droplet radius $r_d = 10\,\mu m$. For the coupling distance we have $R_* = R_i + l_0$ by definition. Assuming $l_0 = 5R_i$ as is done in our air parcel simulations and a typical ice crystal radius $R_i = 100\,\mu m$, we arrive at $R_* = 6R_i = 600\,\mu m$ being reasonably larger than the value for $r_E$.

**Page 6, line 27** Using the exchange rate $J_d/V$ "directly" cannot amount to anything since $V$ is still to be defined. I think the explanation at this point could be improved. $J_d$ is a flux, if we assume that the shell around the ice crystal remains well mixed, then naturally the change in concentration is the flux into the shell divided by the volume of the shell and if there are multiple fluxes they should be summed (which here reduces to multiplying by $N_d$). My suggestion is to eliminate both $V$ and $Z$.
We commented further on the role of the factor $Z$ as a scaling factor after its definition in equation (9). Additionally, we emphasized in section 2.4 that the factor $\frac{1}{ZV}$ can be viewed as a single fitting parameter, but unfortunately, this point of view gives no hint for its explanation or how to assign a value. We believe that our presentation helps in interpreting this factor and suggesting values.

**Page 8, line 2** Please explain why $n$ cannot be estimated from the diffusion constant of water vapour. Effectively $n$ seems to be a fitting parameter for the model, which is fine, but it would be helpful to know why $n$ and not some other quantity should be a free parameter.
Similarly to the response above, we can view the whole factor $\frac{1}{ZV}$ as a fitting parameter which also contains our additional parameter $n$. In our opinion, it is easier to interpret the fitting factor $\frac{1}{ZV}$ as a combination of

two parameters, where one of them is the volume $V$ containing the final fitting parameter $n$. Despite our interpretation, it seems that this fitting parameter $n$ is in some sense artificial, so it cannot be deduced from other physical parameters or constants.

**Page 27** I think the conclusion is much too long, because a large number of observations are repeated. I suggest stressing the most important one or two observations, discussing the prospects for future work and not taking more than half of a page!
We reduced the text in the conclusion section by introducing a table showing the observed effects and trends as suggested by referee #1.

**3.3 Response to Minor Remarks**

**Page 1, line 25** "it may evaporate nearby droplets and grow at their expense". Please rephrase, one cannot use evaporate like this. Perhaps "... may accelerate the evaporation of nearby droplets by growing at their expense".
We followed the suggestion.

**Page 2** What is "ice saturation" please?
Ice saturation indicates that the partial pressure for water vapor equals the saturation vapor pressure over a plane ice surface.

**Page 2** Figure 1 should be made far more compact.
We chose this format for the figure to ease the comparison with figure 1 in Korolev (2007), where the author explained in a similar format the mechanism of the Wegener-Bergeron-Findeisen process.

**Page 4, line 1** What are "objective estimates", perhaps measurement based ...
We indeed thought of estimates based on measurements, but this sentence is deleted in the new version.

**Page 4, line 2** about -> around
We replaced "about" by "around" throughout the manuscript for similar occurrences.

**Page 4, line 12** I suggest avoiding bloat/bloated in physical science writing, perhaps "expand" or "dilate" would be suitable.
We rewrote the paragraphs and avoided those words, see the general response.

**Page 7** eq15 seems to be a trivial rearrangement. I think it is safe to assume that readers can calculate the volumes of spheres and spherical shells.

This rearrangement of the formula is intended in order to show the reader how the factor $\frac{J_d}{ZV}$ is split into two factors.

**Page 9** Figure 4 is superfluous, I suggest just writing a perfectly regular distribution of droplets at the vertices of a cubic lattice.
We followed the suggestion and deleted the figure. Instead of the figure, we included the suggested textual modifications.

**Page 14, line 14** A slightly more detailed description of the parcel model would help this referee. I think we are talking about a parcel of gas moving in the overall flow field.
Parcel models are widely used in cloud physics as a first approximation to cloud formation in reality due to an ascending air volume. The basic concept is an air volume, containing some amount of dry air and water vapor. To apply the equations from thermodynamics for an adiabatic process, the air parcel is thought of as a closed system. It is a good approximation to assume the atmosphere in hydrostatic equilibrium, yielding an equation for the pressure change of the air parcel, see equation (38). As the air parcel moves vertically, its temperature changes adiabatically, see equation (40). In our case, we also assume the air parcel to contain hydrometeors, so we consider the (integrated) variables $q_i$, $q_l$ and $q_v$ for ice, droplets and water vapor. Since the air parcel is assumed as thermodynamically closed, equation (48) follows from the mass continuity equation and allows to model phase changes. The air parcel concept is appealing because it yields only a system of ordinary differential equations instead of partial differential equations but at the same time provides a good first explanation for cloud formation. More detailed explanations about the air parcel concept may be found in Pruppacher and Klett (1997, Chapter 12).

**Page 14, line 22** It is not clear to me exactly what $w$ is. Is it the vertical velocity of the atmospheric flow field at the centre of the air parcel?
The variable $w$ is the vertical velocity of the air parcel, which can indeed be thought of as the vertical velocity of the atmospheric flow field at the centre of the air parcel, i.e. the air volume.

**Page 21, line 12** "get subsaturated", better style would be "become subsaturated". This comment applies throughout the manuscript.
We corrected the wording throughout the manuscript.

**Page 21, line 21** "exists" -> exist
We corrected the mistake.

**Page 21, line 32** "droplet"-> droplets

We corrected the mistake.

**References**

FLEISHAUER, R. P., LARSON, V. E., and HAAR, T. H. V. (2002). "Observed Microphysical Structure of Midlevel, Mixed-Phase Clouds". In: *Journal of the Atmospheric Sciences* 59.11, pp. 1779–1804.

KOROLEV, A. V. (2007). "Limitations of the Wegener-Bergeron-Findeisen Mechanism in the Evolution of Mixed-Phase Clouds". In: *Journal of the Atmospheric Sciences* 64.9, pp. 3372–3375.

KOROLEV, A. V., ISAAC, G. A., COBER, S. G., STRAPP, J. W., and HALLETT, J. (2003). "Microphysical characterization of mixed-phase clouds". In: *Quarterly Journal of the Royal Meteorological Society* 129.587, pp. 39–65.

LEBEDEV, V. (1976). "Quadratures on a sphere". In: *USSR Computational Mathematics and Mathematical Physics* 16.2, pp. 10–24.

PRUPPACHER, H. R. and KLETT, J. D. (1997). *Microphysics of Clouds and Precipitation.* Second revised and enlarged edition. Vol. 18. Atmospheric and Oceanographic Sciences Library. Dordrecht: Kluwer Academic Publishers.

---

## Author Response (AR2)

**Response to referee comments**

Manuel Baumgartner[1,2] and Peter Spichtinger[1]

[1] *Institute for Atmospheric Physics, Johannes Gutenberg University, Mainz, Germany*
[2] *Zentrum für Datenverarbeitung, Johannes Gutenberg University, Mainz, Germany*

January 09, 2018

**1 Response to Comments**

We thank the anonymous referee #2 for his or her time and effort to review our manuscript again and we appreciate the comment. In this response, we copied the comment in blue color and included a marked-up manuscript version.

- The manuscript is now easier to read and understand, from this point of view I think it is ready for publication. The authors disagree with my specific comments (Page 6 line 27 and Page 8 line 2) concerning $ZV$ and later $n$. I think their statement on the first two lines of p9 of the revised manuscript is telling: "With this definition, only a fraction $1/n^3$ of the released water vapor from the droplet will affect the coupling value inside the artificial spherical shell." I agree with this sentence and see my contention that the introduction of $V$ is a complication that makes the mathematical model harder to understand and interpret confirmed by the authors themselves. It they would just confine themselves to having a free parameter between 0 and 1 ($1/n^3$) and using this sentence as a definition I think their work would be clearer. In my view mathematical modelling should be done with a minimum of free parameters and in particular artificial algebraic relations between parameters are very unhelpful, because they obscure model structure by creating the illusion of additional dimensionless quantities, which are rightly a fundamental tool for understanding models in fluid dynamics and thermodynamics.

  It is indeed true that the factor $\frac{1}{n^3}$ amounts to a fitting parameter. In order to make this aspect clearer to the reader, we added another sentence emphasizing this fact and therefore following the suggestion of the referee and the editor.

[revised manuscript text omitted]